# A self-powered intracardiac pacemaker in swine model

Zhuo Liu[1,2,10], Yiran Hu[3,4,10], Xuecheng Qu [1,5,10], Ying Liu[1,5,10], Sijing Cheng[3], Zhengmin Zhang [6], Yizhu Shan[1], Ruizeng Luo[1], Sixian Weng[3], Hui Li[7], Hongxia Niu[3], Min Gu[3], Yan Yao [8], Bojing Shi[1,2], Ningning Wang[6] ✉, Wei Hua[3] ✉, Zhou Li [1,5] ✉ & Zhong Lin Wang [1,9]

Harvesting biomechanical energy from cardiac motion is an attractive power source for implantable bioelectronic devices. Here, we report a battery-free, transcatheter, self-powered intracardiac pacemaker based on the coupled effect of triboelectrification and electrostatic induction for the treatment of arrhythmia in large animal models. We show that the capsule-shaped device (1.75 g, 1.52 cc) can be integrated with a delivery catheter for implanting in the right ventricle of a swine through the intravenous route, which effectively converts cardiac motion energy to electricity and maintains endocardial pacing function during the three-week follow-up period. We measure in vivo open circuit voltage and short circuit current of the self-powered intracardiac pacemaker of about 6.0 V and 0.2 μA, respectively. This approach exhibits up-to-date progress in self-powered medical devices and it may overcome the inherent energy shortcomings of implantable pacemakers and other bioelectronic devices for therapy and sensing.

Implantable bioelectronic devices play an increasingly important role in disease prevention, monitoring, and treatment[1–3], and it is one of the most rapidly emerging fields in medicine[4–6]. As a representative of bioelectronics therapy, the cardiac pacemaker is a powerful tool for bradycardia and heart block therapies[7]. With the upgrading and iteration of technology, pacemakers are becoming intelligent and multifunctional[8–11]. Nevertheless, conventional pacemakers are associated with several complications, such as lead insulation breaks, pocket hematoma, local bulges, and scarring of the skin[12]. Moreover, some patients cannot be implanted with conventional pacemakers due to their comorbidities and venous system defects. To address these lead- and device-pocket-related issues, leadless pacing technology has been proposed. A Leadless pacemaker[13] integrating pacing leads with a pulse generator is very small and can be implanted into the cardiac chamber only through a catheter, which not only reduces pain, and prevents trauma and related complications, but also decreases the risk of infection. Additionally, patients can hardly feel the presence of the pacemaker after implantation, which greatly improves their quality of life.

[1]Beijing Key Laboratory of Micro-nano Energy and Sensor, Beijing Institute of Nanoenergy and Nanosystems, Chinese Academy of Sciences, 101400 Beijing, China. [2]Key Laboratory of Biomechanics and Mechanobiology, Ministry of Education, Beijing Advanced Innovation Center for Biomedical Engineering, School of Engineering Medicine, Beihang University, 100191 Beijing, China. [3]Department of Cardiology, The Cardiac Arrhythmia Center, State Key Laboratory of Cardiovascular Disease, National Clinical Research Center of Cardiovascular Diseases, Fuwai Hospital, National Center for Cardiovascular Diseases, Chinese Academy of Medical Sciences and Peking Union Medical College, 100037 Beijing, China. [4]Department of Cardiology and Macrovascular Disease, Beijing Tiantan Hospital, Capital Medical University, 100070 Beijing, China. [5]School of Nanoscience and Engineering, University of Chinese Academy of Sciences, 100049 Beijing, China. [6]School of Electronics and Information, Hangzhou Dianzi University, 310018 Hangzhou, China. [7]Department of Ultrasound, State Key Laboratory of Cardiovascular Disease, National Clinical Research Center of Cardiovascular Diseases, Fuwai Hospital, National Center for Cardiovascular Diseases, Chinese Academy of Medical Sciences and Peking Union Medical College, 100037 Beijing, China. [8]Department of Cardiology, Beijing Anzhen Hospital, Capital Medical University, 100029 Beijing, China. [9]Georgia Institute of Technology, Atlanta, GA 30332–0245, USA. [10]These authors contributed equally: Zhuo Liu, Yiran Hu, Xuecheng Qu, Ying Liu. ✉e-mail: ning.wang@hdu.edu.cn; drhuawei@fuwai.com; zli@binn.cas.cn

However, it should be noted that the application of implantable bioelectronic devices, including leadless pacemakers, faces various challenges related to power supply in long-term operation[14,15] due to battery capacity limitations[16–18]. Besides, leadless pacemakers are costly and difficult to remove after implantation. Meanwhile, it is difficult to wirelessly recharge a leadless pacemaker because cardiac chambers located in the mediastinum are flooded with blood. In recent years, technologies based on electromagnetic[19–21] or piezoelectric effects[22–27] for converting biomechanical energy into electrical energy have been proposed. For electromagnetic technology, the permanent magnet is heavy and not suitable for magnetic resonance imaging (MRI) examinations[28], which may affect the normal physiological beating of the heart. Its performance is also limited by the beating frequency of the heart[29]. For piezoelectric technology, due to the inherent strength of the heartbeat, the output voltage of piezoelectric devices is generally difficult to meet the pacing threshold requirements[30]. Therefore, it is necessary to develop new energy supply strategies for leadless pacemakers. As a new generation of biomechanical energy harvesting technology, triboelectric nanogenerator[31–34] provide an effective solution for a leadless pacemaker. Currently, triboelectric nanogenerators with different structures and materials have successfully converted biomechanical energy into electrical energy[35–37].

Herein, we proposed a self-powered intracardiac pacemaker (SICP) with a capsule structure for harvesting biomechanical energy from cardiac motion based on the nanogenerator technology. The device can be placed in the right ventricle through an intravenous route by a delivery catheter. The SICP integrates the energy harvesting unit (EHU) and power management unit (PMU) & pacemaker module (PM). In a laboratory experiment, the open-circuit voltage ($V_{oc}$), short-circuit current ($I_{sc}$), and short-circuit charge ($Q_{sc}$) of the device were about 21.8 V, 0.25 μA and 6.4 nC, respectively. We demonstrate that SICP can recharge its PMU by EHU. The material and structural design creates a lightweight, miniature device that maintains stable energy harvesting performance and excellent biocompatibility in vivo. Testing in swine models showed capabilities in the treatment of arrhythmia. Taken together, this work provides an alternative strategy for harvesting biomechanical energy via a minimally invasive approach, which may effectively improve the service life of leadless pacemakers. These findings show that SICP lays the energy foundation for the development of next-generation implantable bioelectronics.

## Results

### Design features and materials of SICP

The schematic illustration in Fig. 1a shows a battery-free and transcatheter SICP in the right ventricle. As shown in the partially enlarged drawing, this capsule-shaped device consists of EHU, PMU&PM, hooks, and radiopaque markers, which can be integrated with a customized delivery catheter system for interventional implantation in the heart through the intravenous route. The SICP fixed on the right ventricular endocardium through the design of the hook structure converts biomechanical energy from cardiac motion to electricity. Figure 1b displays the internal perspective structure of the device. The resin shell of SICP is constructed by stereolithography, the surface of which is deposited Parylene-C by the parylene coating system to form a waterproof coating with good biocompatibility (Supplementary Fig. 1). Polyformaldehyde (POM) pellets, and polytetrafluoroethylene (PTFE) film deposited gold electrodes are employed for the fabrication of the EHU. The POM pellets roll back and forth between the two electrodes under heart beating, which produces an alternating current to the external based on contact electrification (CE) and electrostatic induction. The illustration in Fig. 1c shows CE between POM and PTFE materials can be presented using the surface state model[38]. Material surface electron transfer is the main cause of CE[39]. Based on the CE theory, the basic theory of triboelectric nanogenerator was further

proposed by introducing the displacement current ($\frac{\partial P_s}{\partial t}$) term into Maxwell's equation, where the polarization density $P_s$ is mainly due to the presence of surface electrostatic charges caused by CE. The total displacement current from Maxwell's equation is stated as follows:

$$J_D = \frac{\partial D'}{\partial t} + \frac{\partial P_s}{\partial t} = \varepsilon \frac{\partial E}{\partial t} + \frac{\partial P_s}{\partial t} \qquad (1)$$

where $J$ denotes the density of free conduction current density, $D$ is the electric displacement vector. $E$ represents the electric field, and $\frac{\partial D'}{\partial t}$ is the displacement current due to time variation of the electric field. As the theoretical origin of the triboelectric nanogenerator, $\frac{\partial P_s}{\partial t}$ represents displacement current due to the movement of the changed media as driven by an external mechanical agitation or force. This term is the key to converting mechanical energy into electrical energy in the nanogenerator.

As a three-dimensional explosive view of SICP, Fig. 1d shows various unit components of the device. The integration of PMU&PM achieved the miniaturization for placing in the cylindrical cavity with an outer diameter of 6.8 mm (inner diameter of 5.8 mm) and is, respectively, connected to the tail-end energy harvesting unit and the head-end pacing electrode (Fig. 1e). The overall length of the device was 42 mm and the volume was only 1.52 cc. To prevent reflecting the normal physiological contraction of the heart, the device combined with the heart should typically be <1–2% of the weight of the heart (<3–6 g g)[40]. The overall weight of our device was only 1.75 g, which was better than the above indicators. Besides, scanning electron microscope (SEM) images of POM and PTFE before treatment by inductively coupled plasma (ICP) are shown in Supplementary Fig. 2 and Fig. 1f, respectively. The micro-nano structure on the pellet and the PTFE surface was constructed by the inductively coupled plasma technology for increasing the contact area between the pellet and the surface of the PTFE film during the movement, which will effectively enhance the electrical output performance of the device.

### Working principle and electrical output performance of EHM

To understand the movement process of POM pellets more intuitively, a high-speed camera was employed to capture pellets movement trajectory, which clearly shows that pellets move sequentially from one end to the other on the side wall under the excitation of the external mechanical motion (Fig. 2a). The working principle of EHU is shown in Fig. 2b and Supplementary Movie 1. Under the excitation of cardiac motion, POM pellets roll back and forth on the PTFE surface. After multiple cycles of contact with POM pellets, the PTFE film was negatively charged. Due to the electret properties of the PTFE film, charges can remain on the surface. When POM pellets roll to the left, negative charges are induced on the left electrode. Subsequently, the current is generated in the loop and flows to the left electrode with POM pellets rolling back. With the periodic beating of the heart, under the combined action of gravity and supporting force, POM pellets periodically roll on the curved surface of the PTFE film to generate an alternating current based on the freestanding mode of the triboelectric nanogenerator. A triboelectric nanogenerator can be considered as a capacitor and an ideal voltage source in series. Therefore, the differential equation of triboelectric nanogenerator with an external pure resistance R can be represented by Kirchhoff's law[41] as follows:

$$R\frac{dQ}{dt} = -\frac{1}{C}Q + V_{oc} \qquad (2)$$

Where $Q$ denotes the transferred charge, $C$ is the triboelectric nanogenerator's internal capacitance, and $V_{oc}$ represents the open-circuit voltage.

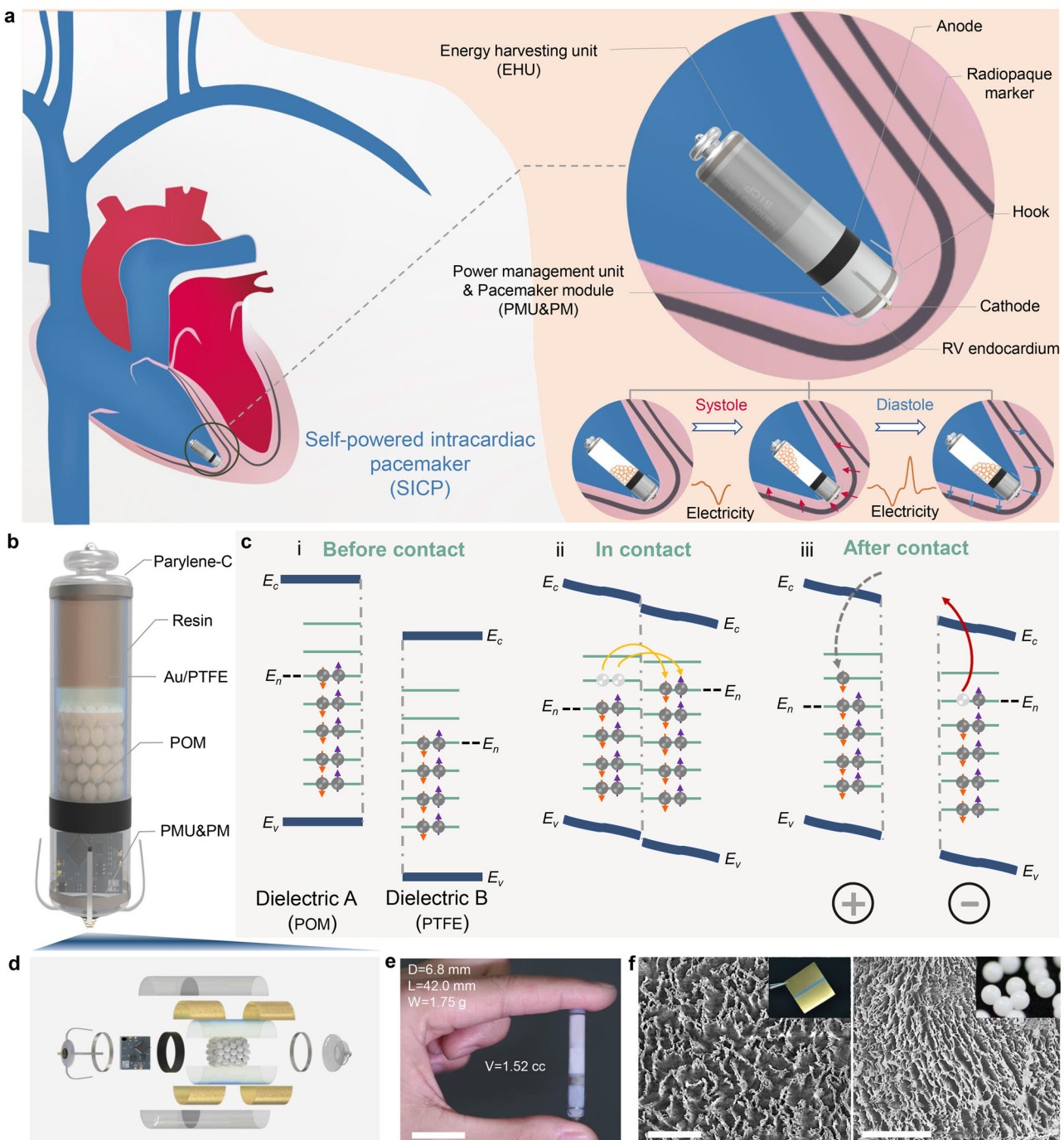

**Fig. 1 | Design features and materials of a self-powered intracardiac pacemaker (SICP). a** Schematic illustration of the device in the right ventricle (RV) for converting biomechanical energy from cardiac motion to electricity and regulating arrhythmia, which is composed of an energy harvesting unit (EHU), a power management unit & pacemaker module (PMU&PM), hooks and radiopaque markers. **b** Internal perspective structure of the device. **c** Charge transfer (i) before contact, (ii) in contact, and (iii) after contact between two different dielectrics-

polyformaldehyde (POM) and polytetrafluoroethylene (PTFE). **d** Three-dimensional explosive view of SICP. **e** Photograph of SICP with a weight of 1.75 g and volume of 1.52 cc (diameter = 6.8 mm; length = 42 mm) (Scale bar = 2 cm). **f** Scanning electron microscope (SEM) images of the surface of POM pellet and PTFE film after treatment by inductively coupled plasma (ICP) technology (Scale bar = 5 μm). The abbreviations are the same in other figures unless otherwise stated.

The open-circuit voltage is proportional to the transferred charge. Furthermore, the short-circuit current $I$ can be represented as follows:

$$I = \frac{dQ}{dt} = \frac{dQ}{dx}\frac{dx}{dt} = \frac{dQ}{dx}v \qquad (3)$$

The short-circuit current also relies on the external motion velocity (movement frequency). We compared the electrical performance

of EHU at different numbers of pellets and tilt angles. As shown in Fig. 2c, for the volume ratio ($V_{pellets}/V_{cavity}$) from 10% to 90%, the maximum value of the $V_{oc}$ was about 21.8 V at 40%. It was found that the overall height of the globules under a 40% volume ratio is consistent with the height of the unilateral electrode, which enables a maximum effective contact area during movement. The motion direction is consistent with the direction of the long axis of the device which was defined as 0° (Supplementary Fig. 3). The electrical output

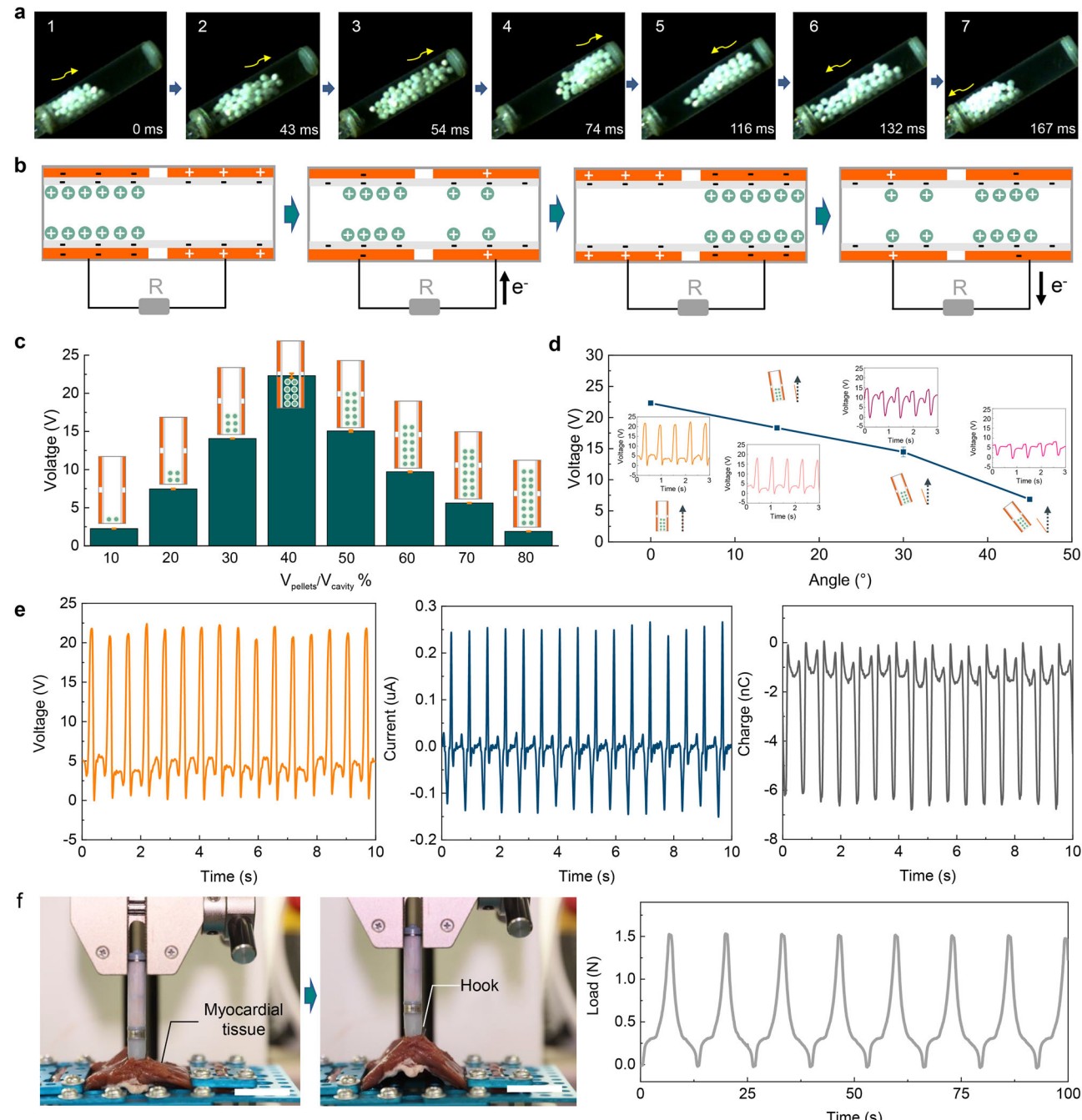

**Fig. 2 | Working principle and electrical output performance of energy harvesting unit (EHU). a** Photographs of the movement process of the pellets in EHU. **b** Schematic diagram of the working principle of EHU. **c** Open-circuit voltage ($V_{oc}$) of EHU when working under different volumes of pellets. ($n = 3$, Data are presented as mean ± SD) **d** $V_{oc}$ of EHU when working under different tilt angles. ($n = 3$, Data are presented as mean ± SD) **e** $V_{oc}$, short-circuit current ($I_{sc}$), and short-circuit charge ($Q_{sc}$) of EHU in vitro. **f** Tensile test for SICP (Scale bar = 2 cm). Source data are provided as a Source Data file.

performance of the device gradually decreases with an increase in the angle (Fig. 2d). The highest performance of EHU was obtained at the angle of 0°. More importantly, when the tilt angle was 30°, the output voltage of the device was maintained at >65% of the maximum output. The $V_{oc}$ of EHU could still reach 6.0 V when the tilt angle reached 45°. The good compliance of the device to the angle is mainly due to the design of the spherical structure for POM materials. As shown in Fig. 2e, under the optimal motion angle and volume ratio, the $V_{oc}$, $I_{sc}$, and $Q_{sc}$ of the device were about 21.8 V, 0.25 μA, and 6.4 nC, respectively. The excellent fixation effect of the device is a key factor in ensuring good electrical output in vivo. In addition, the tensile test was employed for SICP. As shown in Fig. 2f, the device can withstand a

cyclic action of 1.5 N tensile force (Supplementary Movie 2). The isolated heart simulation experiment also shows that SICP achieves good fixation in the heart (Supplementary Fig. 4 and Supplementary Movie 3). The $V_{oc}$ and $I_{sc}$ of EHU show opposite trends with different load resistance (Supplementary Fig. 5a). The EHU reached a maximum output power density of 2200 mW/m³ at 100 MΩ (Supplementary Fig. 5b). After 6 million stimuli cycles by a linear motor, the $V_{oc}$ of EHU maintained stable in compared with its initial state (Supplementary Fig. 6), exhibiting outstanding stability and durability, which enables for long-term harvesting biomechanical energy in vivo. In addition, the output performance of EHU was further improved when the motion frequency was increased within a certain range (Supplementary Fig. 7).

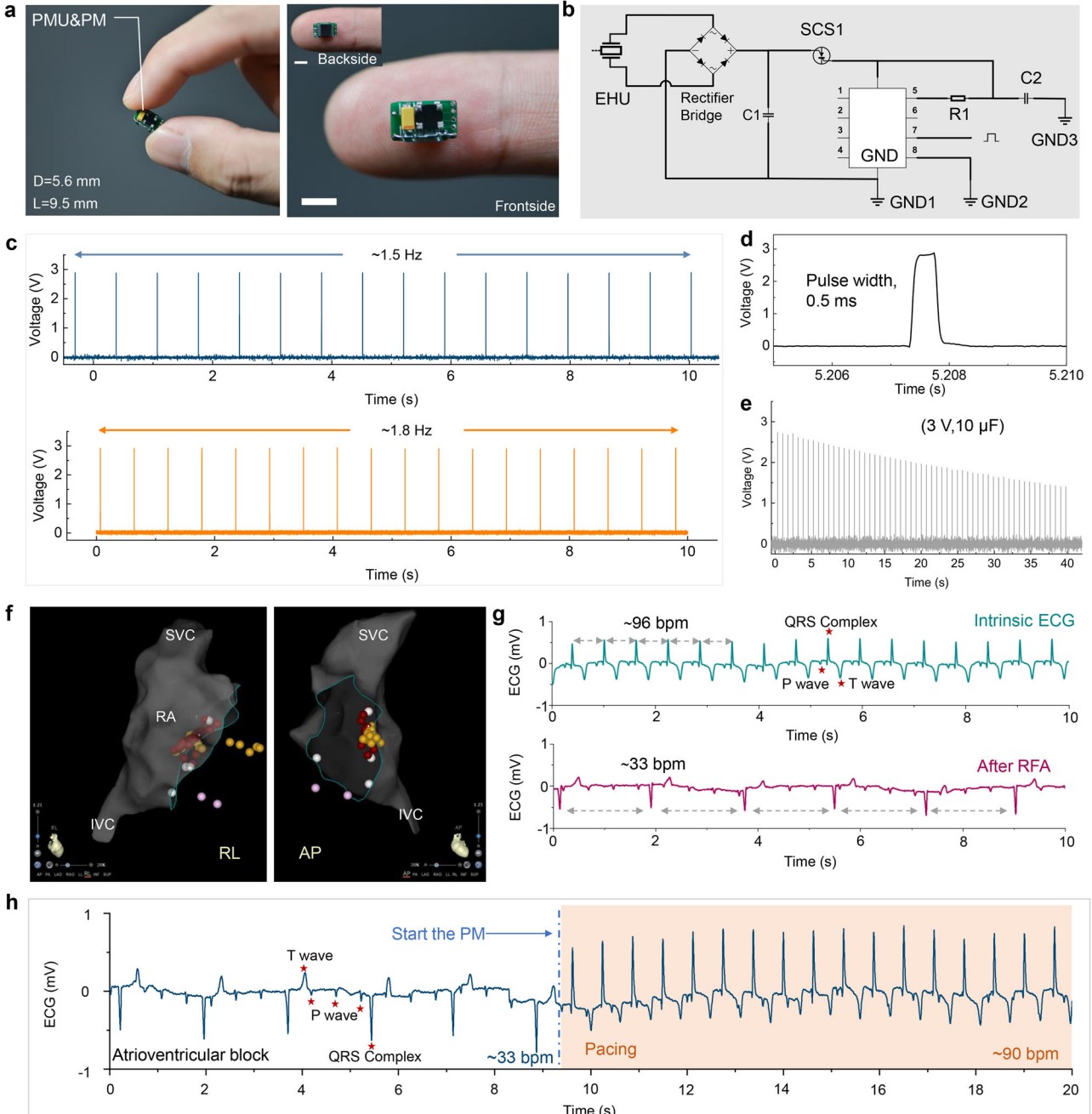

**Fig. 3 | Characterizations for power management unit & pacemaker module (PMU&PM) of SICP. a** Photographs of the integrated PMU&PM (Scale bar = 5 mm). **b** Circuit diagram of PMU&PM. **c** Stimulation pulse generated by PM with different frequencies. **d** Enlarged view of the stimulation pulse. **e** PM powered by the capacitor (10 μF, 3 V) that continuously emits stimulating electrical pulses. **f** Electroanatomic reconstructions of the cardiac structures and ablation target (red dots) by three-dimensional mapping system (superior vena cava (SVC); right atrial (RA); inferior vena cava (IVC); anterior-posterior (AP), right-left (RL) projections). **g** Electrocardiogram (ECG) before and after construction of an AVB animal model. **h** ECG of an atrioventricular block (AVB) animal model before and after pacing by PM with a driving voltage of 1.5 V. Source data are provided as a Source Data file.

## Characterizations for PMU&PM

We developed an integral power management unit & pacemaker module (PMU&PM) (Fig. 3a). The length and width of a PMU&PM were 9.5 mm and 5.6 mm, respectively. The PMU&PM consisted of a rectifier bridge, capacitor, reed switch, electric pulse chip, and peripheral circuit (Fig. 3b). The extremely narrow width design enables PMU&PM to be integrated with EHU for establishing SICP. The pulse frequency of PM can be modulated on demand (Fig. 3c), such as 1.5 Hz, 1.8 Hz, etc., which induces myocardial contraction and regulates heart rate via pacing electrodes. Here, the output voltage and pulse width of the

electrical pulses were set to 3 V and 0.5 ms (Fig. 3d), respectively. Charging a 10 μF capacitor to 3 V by EHU can continuously power the PM to work for nearly 40 s (Fig. 3e). Moreover, we also verified that using a capacitor with a larger capacity (47 μF) can significantly increase the duration of the pacing pulse release for the PM (Supplementary Fig. 8). To further explore the pacing efficiency for PM in animals, complete atrioventricular block (AVB) was induced by radiofrequency ablation (Fig. 3f). Figure 3g shows the electrocardiogram (ECG) before and after atrioventricular node ablation in a swine model. The heart rate decreased from 96 bpm to 33 bpm, showing that the

bradycardia animal model was built successfully. The PM with 1.5 Hz was used for pacing the AVB animal model. The driving voltage of the pacing chip was increased to 1.5 V, and effective pacing in AVB animals was achieved (Fig. 3h), which is consistent with the threshold voltage of the PM of 1.5 V in large-animal models.

Biocompatibility is a critical aspect of the success of implantable bioelectronic devices, which depends on the encapsulation strategy and materials. Before implanting the device, we systematically evaluated the biocompatibility of the encapsulation material of SICP. The cytoskeletal structures and cell nucleus were detected by immunofluorescence staining on days 1, 2, and 3, respectively (Supplementary Fig. 9). The viability of a fibroblast cell line L929 on the encapsulation film was tested by the Cell Counting Kit-8 (CCK-8) (Supplementary Fig. 10). Compared with the control group, the results revealed that encapsulation material had no negative impact on cells growth and proliferation. Meanwhile, localized tissues of the skin to deep layer muscle from the implantation location of the materials were stained by Hematoxylin and Eosin (H&E). The tissues surrounding the materials of the device showed no observable differences compared with the surrounding tissue (Supplementary Fig. 11). Moreover, acceptable blood compatibility was required for the encapsulation materials in the present study. Hemolysis and coagulation on the materials were also demonstrated. The average hemolysis rate of the encapsulation materials was remarkably lower than the National Standards Organization (ISO) standard (5%) (Supplementary Fig. 12a). Platelets on the material that maintained a round shape without obvious deformation and aggregation, indicating a low degree of activation (Supplementary Fig. 12b).

### Energy harvesting of SICP in vivo

To simulate the function of SICP for biomedical energy harvesting from cardiac chambers in human bodies, adult swine (age: 1.2–2.0 years; weight: 50–65 kg) were selected as our animal model. After swine were anesthetized and intubated with respirators for artificial respiration, ECG and femoral artery pressure were recorded using the data acquisition hardware (MP150, BIOPAC System, Inc.). The right-side neck skin was prepared by tincture of iodine solution, and the external jugular vein was exposed with a small incision (Supplementary Fig. 13). The diameter of the device was smaller than that of the external jugular vein, indicating that the device could be successfully delivery into the right ventricle through the venous system (Supplementary Fig. 14).

Once the vein access was obtained by the homemade introducer and dilator advancement over a guidewire (RF*GA35153M: Terumo Corporation, Tokyo, Japan), the homemade delivery catheter by integrating SICP was advanced across the tricuspid valve into the right ventricle, and SICP was delivered into the right ventricular endocardium. Figure 4a shows fluoroscopy images of minimally invasive delivery of SICP to the right ventricle via the intravenous route. The schematic diagram corresponding to the fluoroscopy images is shown in Supplementary Fig. 15. The electrodes and PMU&PM of the device could be seen clearly through dynamic Movie (Supplementary Movie 4 and Supplementary Movie 5). The constructed radiopaque marker could accurately judge that the device successfully reached the right ventricle. The hook on the front end of the device interacts with the endocardium to hold the device firmly on the heart, and the helical electrode (cathode) makes good contact with the endocardium (Supplementary Movie 6).

After implanting the device, no significant changes in ECG and blood pressure of the animal model (Fig. 4b). The delivery process did not affect the physiological state of the experimental animals. The cardiac motion caused the pellets to move back and forth on the surface of the PTFE membrane (Supplementary Movie 7). The EHU of SICP generates periodic alternating current based on a coupling effect of triboelectrification and electrostatic induction. $V_{oc}$, $I_{sc}$, and $Q_{sc}$ of

SICP were about 6.0 V, 0.25 µA, 8.5 nC in vivo (Fig. 4c). With the periodic physiological contraction and relaxation of the heart, globules move freely in SICP, and the electrical signal exhibits a certain volatility. We speculate that fluctuations in the electrical signal may also be affected by the blood flow. Nonetheless, our statistical analysis showed that the average voltage and current can reach 4 V and 0.2 µA (Fig. 4d), respectively, and the energy conversion efficiency of the voltage exceeding 1.5 V per unit time accounted for over 82%. The output voltage EHU meets the requirement of the threshold voltage of PM. Actually, cardiac contraction intensity is also affected by respiratory status and exercise. Therefore, appropriately enhancing cardiac functional status may be beneficial to improving energy harvesting efficiency.

### Illustration pacing performance of SICP in vivo

The EHU of SICP converts the biomechanical energy from heart beating to electricity, which can be stored in the capacitor of the PMU for powering the PM after the reed switch is turned on by the magnet (Fig. 5a). That is, after the device is integrated with the heart, self-sufficiency of electrical energy is achieved. The voltage of a capacitor was charged from 0 V to 3 V within 9000 s with the same electrical output of SICP in vivo (Supplementary Fig. 16). Corresponding to the ECG and blood pressure, transthoracic echocardiograms also showed that no tricuspid regurgitation was observed after SICP implantation (Fig. 5b). This result further suggests that the physical structure of the device itself has no effect on the heart. To demonstrate pacing the performance of SICP, ECG signals of the swine during SICP operation are shown in Fig. 5c. The typical P wave, QRS complex, and T wave appear in sequence on the intrinsic ECG. When SICP was in operation, the premature-paced QRS complex was induced by the electrical pulse stimulus (Supplementary Movie 8). The ECG showed regular-paced QRS complex occurrence ahead of the P wave (atrial contraction), indicating that the ventricle was effectively captured by SICP. The heart returned to intrinsic rhythm after the PM of SICP stopped working. The above results demonstrate the effectiveness of SICP for pacing the heart rhythm. Furthermore, the wound was sutured after implanting the device into the right ventricle, and long-term experiments were performed. As shown in Fig. 5d, e, we monitored the ECG signals of the experimental animals for 2 and 3 weeks, respectively. The results showed that the heart rhythm remained stable, and the experimental animals were eating and living normally without the occurrence of complications (Supplementary Fig. 17). Meanwhile, EMU was controlled by switching in the third week, and SICP effectively paced the heart in vivo, the heart rate of the animal increased from 90 bpm to 108 bpm, demonstrating that the device still maintained normal pacing function (Fig. 5f). Three weeks after implantation, the swine was sacrificed by injection of a medium with high potassium concentration, and then the chest was opened to remove the heart with SICP. We found that the device was firmly anchored to myocardial tissues (Fig. 5g). Masson's trichrome stain was applied after prolonged implantation, which revealed that the extent of cardiac tissue injury and inflammatory fibrous hyperplasia occurred only at the device fixation site, and Hematoxylin and eosin (H&E) staining showed no detectable infiltration of lymphocytes in other sites, prompting the conclusion that neither humoral nor cellular rejection to the device occurred in the myocardium from the implantation site (Fig. 5h). Overall, these in vivo tests demonstrated that SICP can achieve long-term pacing in large-animal models.

## Discussion

We develop a SICP and demonstrate its efficacy and safety in the cardiac pacing of large-animal models. This device provides a promising method to harvest biomechanical energy from cardiac motion for powering the pacemaker module with the significant advantages of being leadless, battery-free, transcatheter-

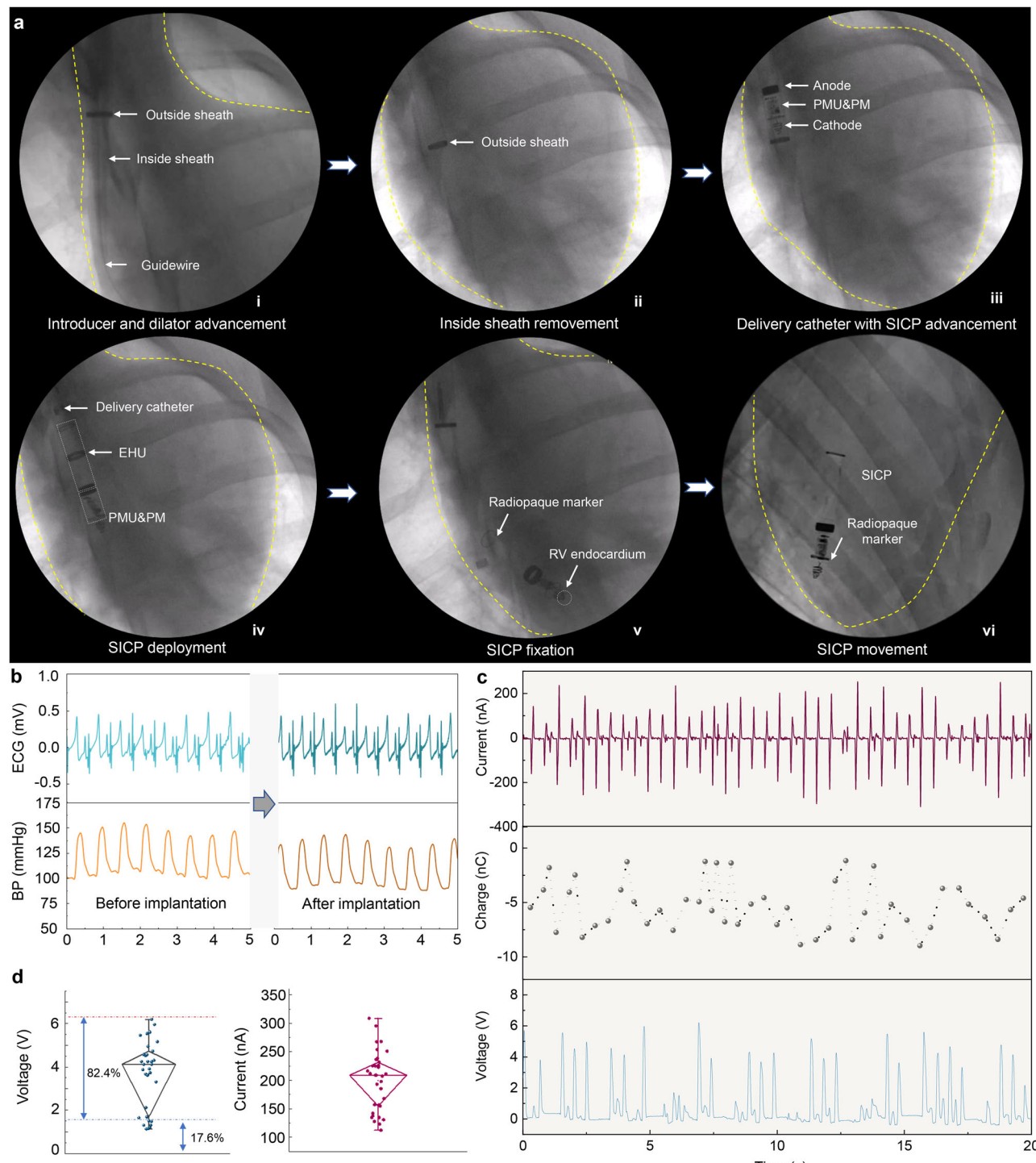

**Fig. 4 | Energy harvesting of SICP in vivo. a** Fluoroscopy images of minimally invasive delivery of SICP to the right ventricle via the venous system under fluoroscopic guidance. **b** ECG and blood pressure signals of the animal before and after SICP implantation. **c** $I_{sc}$, $Q_{sc}$, and $V_{oc}$ of SICP in vivo. **d** Statistical analysis of $V_{oc}$ and $I_{sc}$ (there are 34 data points for each test group. The box ends represent the 25th and 75th percentiles. The horizontal line in each box represents the median. The upper and lower whiskers refer to the range of non-outlier data values. Outliers were plotted as individual points). Source data are provided as a Source Data file.

intervention, and lightweight. Based on the synergy between EHU and cardiac motion, the constant supply of energy enables the pacemaker to work stably, thereby preventing the perioperative risk caused by the replacement of devices due to energy depletion. In addition, to ensure cardiac physiological activity, the overall device adopts lightweight materials to reduce the load on the heart. Minimally invasive intervention with delivery technology decreases the risk of surgical exogenous infection and tissue trauma. The capsule

structure facilitates implantation through the venous system. Such a capsule structure also substantially improved the energy conversion efficiency for EHU in vivo based on the freestanding triboelectric-layer mode of the triboelectric nanogenerator. The biomechanical energy harvested from SICP from each cardiac cycle is about 0.026 μJ (Supplementary Note 1). Maximum power output of SICP is about 0.039 μW (Supplementary Note 2). Theoretically, it means that the energy harvested by SICP from four heartbeats will be

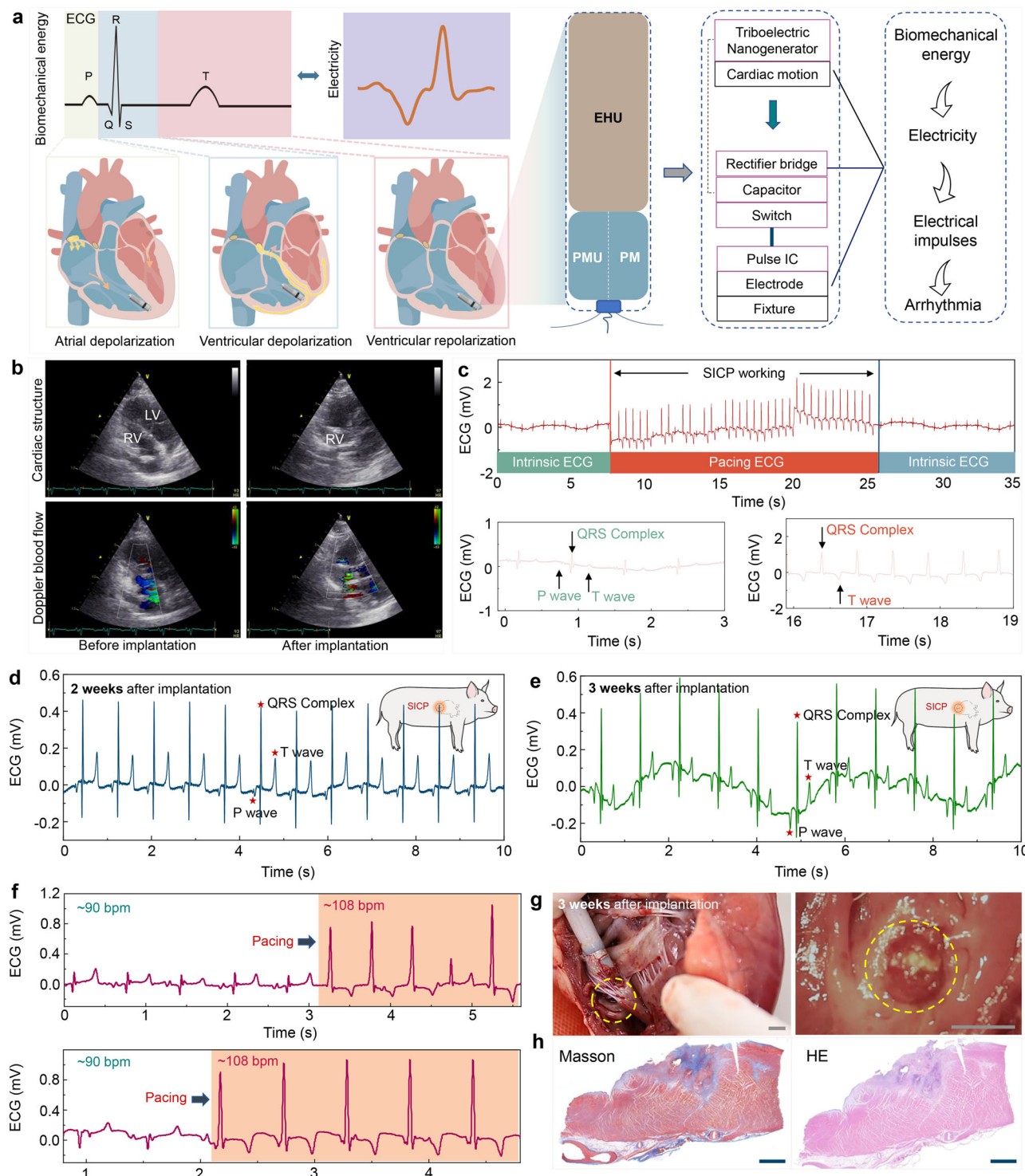

**Fig. 5 | Illustration pacing performance of SICP in vivo. a** Block diagram of the components of SICP. **b** Transthoracic echocardiograms of SICP before and after implantation. **c** Animal ECG signal when SICP works during minimally invasive implantation. **d** Animal ECG signal of 2 weeks and **e** 3 weeks after implantation. **f** Animal ECG signal when SICP works at 3 weeks. **g** Photographs of SICP fixed to the endocardium at 3 weeks (Scale bar = 0.5 cm). **h** Photographs of Masson's trichrome and Hematoxylin−eosin (HE) staining for myocardial tissue at the fixation site (Scale bar = 200 μm). Source data are provided as a Source Data file.

higher than the pacing threshold energy of a commercial leadless cardiac pacemaker (Supplementary Note 1).

Meanwhile, EHU is mainly constructed of polymer materials, which provides feasibility for SICP to be compatible with MRI examinations during clinical applications. Specifically, SICP has good blood and tissue compatibility and does not cause significant inflammation in the endocardium. Three weeks after the operation, the experimental

animals maintained a normal survival state, and the device exhibited excellent output performance. Large-animal experimental models effectively simulate clinical applications and may provide more valuable and comparable results. Although SICP has certain limitations in long-term constant pacing on clinical criterion, this work provides a proof-of-concept demonstration for the next-generation pacemaker and will facilitate the upgrade of existing commercial leadless

pacemakers (Table S1). Furthermore, with the in-depth follow-up research and the improvement of the efficiency of the EHU, we believe that the energy collected by SICP from one heartbeat can fully satisfy the leadless pacemaker for one pacing. Going forward, further research is required to investigate the self-powered closed-loop operation system integrating arrhythmia active monitoring and stimulation regulation.

## Methods

### Fabrication of SICP
The enclosure of SICP was printed by a 3D printer (PHTON MONO) using UV-sensitive resins (ANYCUBIC Photon). Nanostructured PTFE film (50 μm) and POM pellets (diameter: 1.588 mm) were employed as triboelectric layers, processed by an inductively coupled plasma etching system (SENTECH/SI 500). In detail, the PTFE films and POM pellets were rinsed with alcohol and deionized water. The Au, which acted as the mask for the etching process, was sputtered onto their surfaces for about 30 s. Then the PTFE films and POM pellets were etched by ICP reactive ionic etching for 300 s (ICP power: 400 W and 100 W) and 150 s (ICP power: 500 W and 150 W), respectively. The reaction gas in the ICP process was CF4 (30.0 sccm), $O_2$ (10.0 sccm), and Ar (15.0 sccm). The two Au electrodes were deposited on the back of nano-PTFE by magnetron sputter (Denton Discovery 635) parallelly for 15 min (sputter power 50 W), the gap between them was about 1 mm. The Au bottom electrodes were polarized by connecting with wire to ground with a voltage of 4.5 kV for 15 min through the corona needle. The composite film was attached to the inner surface of the enclosure. The POM pellets were placed in the cavity of the energy harvesting unit. Altium Designer software was used to design the power management unit and pacemaker module. The custom control chip is an ultra-low power chip, which has the characteristic of a low power supply voltage range. The circuit board adopts the welding process of surface mounted technology (SMT). The spiral platinum-iridium alloy was attached to the bottom of SICP as cathode and the ring of platinum-iridium alloy was attached to the waist of SICP as anode. Narrow strips of tungsten sheet (thickness: 50 μm, width: 1 mm) were rolled into rings and fixed to both ends of SICP as a radiopaque marker. Nickel alloy wires were bent into an arch (diameter: 0.3 mm, length: 10 mm) and crossed through holes in the bottom of SICP as a hook to anchor in the myocardial wall.

### Encapsulation of the SICP
The ethyl cyanoacrylate (Aibida, Guangzhou, China) was employed to close the seams of the enclosure. Then the one-component UV light-curing adhesive (8500 Metal, Switzerland) was spin-coated on the enclosure as the package layer and then cured under UV light for 10 s. The holes in the enclosure that led out the wires need to be sealed with light-curing glue several times. Finally, the parylene-C particles were steamed at 135 °C/690 °C and then deposited on the surface of SICP where the thickness of parylene is 3 μm.

### Cell viability
The L929 cells (fibroblasts, GNM28) were acquired from the Cell Bank of the Chinese Academy of Sciences in Beijing, China. After being cultured to a stable stage, L929 cells were collected and seeded on 96-well tissue culture polystyrenes (TCPs, Corning, USA) with a density of $1 \times 10^6$ cells/ml. The culture dish of the experimental group deposited encapsulation materials. 20 μL CCK-8 solution (Solarbio, China) was mixed with 280 μL cell culture medium for each well. The cells were cultured for 1, 2, and 3 days and evaluate the cell viability every day. After incubating the cells with CCK-8 solution for 1 h at 37 °C, 200 μL cell culture supernatant was transferred into a 96-well plate (set at least three repetitions for each group). The absorbance of the solution in the 96-well plate was measured at 450 nm with a microplate absorbance assay instrument (Bio-rad iMark, USA).

### Cell morphology and immunofluorescent staining
Cell morphology could show the cellular growth condition on different substrates. The cytoskeleton and nucleus were stained with Phalloidin (Abcam, USA) and 4′, 6-diamidino-2-phenylindole (DAPI, Solarbio, China). The L929 were cultured for 1, 2, and 3 days on the naked 96-well disposable confocal dish, and the dish deposited encapsulation materials for cell morphologic observation. Before staining, the cells were rinsed with phosphate buffer solution (PBS, Solarbio, China) three times gently, then fixed with 4% paraformaldehyde (Solarbio, China) for 10 min and permeabilized with 0.1% Triton X-100 (Solarbio, China) for 10 min. Finally, the fixed cells were stained with Phalloidin for 40 min, DAPI for 10 min at room temperature, and washed with PBS three times. The stained cells were visualized by the laser scanning confocal microscope (SP8, Leica, Germany) under the filter at $E_x/E_m = 493/517$ nm.

### Platelet adhesion tests
Rats were anesthetized with 2% isoflurane (RWD, R510-22-4). After the location of the abdominal aorta was determined by laparotomy, 2 ml of fresh blood from arterial blood was collected using a blood collection needle. The collected blood was placed in a centrifuge tube for 30 min. The cells were centrifuged at $110 \times g$ for 10 min in a centrifuge. Platelets were removed and incubated on sterilized encapsulation coating material for 90 min at room temperature. After 30 min of fixation with 4% paraformaldehyde (Solarbio, China), platelets were dehydrated in serial concentration gradients of ethanol (50%, 60%, 70%, 80%, 90%, 95%, and 100%) for 30 min. After evaporation to dryness at room temperature, the surface of the encapsulation coating material with platelets was visualized by SEM.

### Hemolysis assay
Working solution: positive group: 0.3% Triton X-100 (Solarbio, China); Negative group: normal saline (0.9% NaCl); Material group: 1 mg/ml encapsulation coating material leaching solution (1 mg encapsulation coating material was immersed in 1 ml of normal saline for 3 days). 1 ml of fresh blood from male 6-week-old SD rats (220 g) was placed in a 15 ml centrifuge tube, and the supernatant was removed after 4 ml PBS (Solarbio, China) was washed 4–5 times (170 g, 5 min). The washed red blood cells (RBCs) were resuspended by 10 ml PBS (Solarbio, China). 0.2 ml resuspended red blood cells were mixed with 0.8 ml working solution and incubated for 4 h. The mixture was centrifuged in a centrifuge at $170 \times g$ for 5 min and photographed for recording. The supernatant was removed and the OD value (541 nm) was measured by a microplate reader. % hemolysis = $(OD_{test} - OD_{neg})/(OD_{pos} - OD_{neg}) \times 100\%$.

### Histology
The tissues were fixed in a 4% paraformaldehyde (Solarbio, China) overnight at room temperature, followed by dehydration using a series of graded ethanol and xylene solutions (Solarbio, China). Subsequently, routine paraffin embedding was carried out, and tissue sections with a thickness of 4 μm were obtained using a microtome. To visualize the tissue morphology and extracellular matrix, hematoxylin-eosin (HE) and Masson's trichrome staining (Solarbio, China) were performed with standard procedures. The resulting images were observed under a light microscope.

### Animal preparation
All experimental processes were strictly in line with the institutional and national guidelines for the care and use of laboratory animals and the study protocol was reviewed and approved by the Ethical Committee of the Animal Experimental Center in the State Key Laboratory of Cardiovascular Disease and Fuwai Hospital (0103-1-1-ZX(Y)−3). The swine were anesthetized and intubated with respirators for artificial respiration, and then ECG and femoral artery pressure were recorded

by the data acquisition hardware (MP150, BIOPAC System, INC.). Adult swine (age: 1.2–2.0 years; weight: 50–65 kg) were used ($n = 8$, female = 5, male = 3). Rats (male, 220 g, 6 weeks of age, $n = 10$) were purchased from the Beijing Vital River Laboratory Animal Technology Co., Ltd., China.

## Induction of AVB
Complete atrioventricular block (AVB) in the swine model was induced by radiofrequency ablation. The right femoral vein was dissected and cannulated with an 8F sheath to introduce a contact force catheter (Thermocool SmartTouch, Biosense Webster Inc., Diamond Bar, CA). Electroanatomic reconstructions of the cardiac structures (superior vena cava (SVC), inferior vena cava (IVC), and right atrial (RA)) were performed using a three-dimensional mapping system (CARTO3, Biosense Webster, Inc., Diamond Bar, CA). After mapping a (or cluster of) near field His-bundle (HB) potential, radiofrequency ablation with a power of 35 W and a duration of 120 s was performed in this HB region (ablation target). The signs of successful ablation were as follows: (1) the emergence of complete AVB; (2) the occurrence of the escape rhythm. The ablation catheter with the 8F sheath was then withdrawn and the incision of the right femoral vein was sutured.

## Implantation of SICP
First, the right-side neck skin was prepared with a tincture of iodine solution. Then, the external jugular vein was exposed with a small incision. Next, a low-dose bolus of heparin (2000 to 4000 units) was delivered intravenously. After the vein was punctured, the next step was to guide a stiff guidewire (RF*GA35153M: Terumo Corporation, Tokyo, Japan) down into the IVC. Under fluoroscopic guidance, the delivery sheath was advanced through the external jugular vein and into the right atrium (RA) over the guidewire. The radiopaque marker band on the tip of the outside sheath should be positioned in the mid-RA, followed by retracting the inside sheath. Then the homemade delivery catheter integrated with SICP was advanced across the tricuspid valve into the right ventricle via the outside sheath. Once SICP was deployed with adequate fixation, the homemade delivery catheter with the outside sheath was retracted from the external jugular vein leaving SICP in the final position of the right ventricular apex. Upon removal of the sheath, the right-side neck skin and the vein were sutured.

## Characterization methods
The scanning electron microscopy (SEM) images were taken by a Hitachi field emission scanning electron microscope (SU 8020). The $V_{oc}$, $I_{sc}$, and $Q_{sc}$ were detected by an electrometer (Keithley 6517B) and recorded by an oscilloscope (Teledyne LeCroy HDO6104). The tensile test of SICP was performed by the ESM301/Mark-10 system. Optical photographs of the movement of the pellets in the cavity were exhibited by a high-speed digital video camera (FASTCAM Mini AX200 JAPAN). Optical photographs and movies of SICP fixed on the endocardium of the right ventricle in an isolated heart were recorded by an endoscope (BRK pt50).

## Statistical analysis
All experiments were repeated at least three times. Data were analyzed as mean ± standard deviation (SD). The statistical significance of the differences is determined by a tailed $t$-test. ns was considered no significant differences. Origin 2018, GraphPad Prism version 8.0. and Excel were used for data analysis and plotting.

## Reporting summary
Further information on research design is available in the Nature Portfolio Reporting Summary linked to this article.

## Data availability
The authors declare that all data supporting the results of this study are available within the paper and its Supplementary Information. The source data underlying Figs. 2c–f, 3c–e, 3g, 3h, 4b–d, and 5c–f and Supplementary Figs. 5–8, 10, 12, and 16 are provided as a Source Data file (https://doi.org/10.6084/m9.figshare.24715311). Any additional requests for information can be directed to, and will be fulfilled by, the corresponding authors. Source data are provided in this paper.

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

## Acknowledgements

We are grateful to the laboratory members, Prof. Yubo Fan (Beihang University), Dr. Lingling Xu (National Center for Nanoscience and Technology), and Dr. Pengkang He (Peking University First Hospital, Beijing) for their cooperation in this study. This work was financially supported by grants from the National Natural Science Foundation of China (T2125003 to Z.L., 82102231 and 82372141 to Z. Liu, 61875015 to Z.L. and 82100325 to Y.H.), the National Key Research and Development Program of China (2022YFE0111700 to Z.L.), Beijing Natural Science Foundation (JQ20038 to Z.L.), Beijing Natural Science Foundation (L212010 to Z.L.), High-level hospital clinical research funding of Fuwai Hospital, Chinese Academy of Medical Sciences (No.2022-GSP-GG-11 to W.H.), the Fundamental Research Funds for the General Universities to Z. Liu, the China Post-doctoral Science Foundation (2023M731943 and BX20230169 to X.Q.), The Beijing Gold-bridge project (No. ZZ21055 to Y.H.).

## Author contributions

Z. Liu, Y.H., X.Q., and Y.L. contributed equally to this work. Z. Li, W.H. and Z. Liu conceived the idea and guided the project. Z. Liu, Y.H., X.Q., and Y.L. designed the experiment and analyzed the results. Z.L.W. directed the preparation of EHU. Z. Liu, X.Q., Y.L., N.W., Z.Z. and B.S. fabricated SICP and performed the electrical characterization. Y.S., R.L., and Z. Liu performed the cell experiments. Z. Liu, Y.H., X.Q., Y.L., S.C., S.W., H.L., H.N., M.G. and Y.Y. performed in vivo surgery. Z. Liu, Y.H., X.Q. and Y.L. wrote the paper. All authors read and approved the final manuscript.

## Competing interests

The authors declare no competing interests.
