## [Peer review file · Nature Communications]

REVIEWER COMMENTS

Reviewer #1 (Remarks to the Author):

This comprehensive study presents a self-powered intracardiac pacemaker (SICP) that utilizes a nanogenerator to harness biomechanical energy from cardiac motion. The SICP, designed with a capsule structure, can be implanted in the right ventricle using a delivery catheter. It effectively converts cardiac motion energy into electricity, maintaining endocardial pacing function over a three-week follow-up period. The device demonstrates impressive open-circuit voltage and short-circuit current values, showcasing its potential to overcome energy limitations in implantable pacemakers. The SICP's lightweight design, stable energy harvesting performance, and successful testing in swine models highlight its promising role in treating arrhythmia and extending the service life of leadless pacemakers. In the sense that this work represents a significant advancement in the field of implantable bioelectronics, I would recommend this manuscript for possible publication in Nature Communications after minor revision.

Comments:

1. Some important articles about implantable triboelectric nanogenerators and their biomedical applications could be referred to provide readers with more diverse examples of research such as: "doi.org/10.1038/s41467-021-24417-w", "doi.org/10.1126/science.aan3997", "doi.org/10.1002/smt.202201350", "doi.org/10.1002/adma.202209054". Citing these recent articles can provide supplemental information for the phrase 'to power supply in long-term operation due to battery capacity limitation' in line 60.
2. To ensure accurate information, I kindly request a typo correction. In line 156, 'PFEF film' appears to refer to 'PTFE film' in the given context.
3. In Fig.2a, a series of images captured with a high-speed camera showcases the movement process of the POM pellet. It would be helpful to provide information on the total time it took to capture the images as an indication of the time intervals between each photo. This information would contribute to a better understanding of the movement process, considering the driving speed as a factor influencing the POM pellet's movement.
4. In Fig.2d, voltage measurements are presented according to the driving angle of the device. It is observed that all output patterns were driven at the same frequency. It would be beneficial to provide details of the experimental setup used to standardize the applied speed and angle. This information would greatly assist other researchers in conducting follow-up studies.
5. According to the manuscript and Fig.2f, a tensile test was conducted to demonstrate the excellent fixation effect of the device. However, peel-off testing is generally considered a more suitable evaluation method for demonstrating the fixation effect. It would be helpful to know the reasons behind conducting the test as described in the manuscript.
6. Building upon the previous comment, in Fig.2f, only the information on the applied load over time is provided, which seems different from the factors typically associated with demonstrating an excellent fixation effect. It is suggested to calculate values such as adhesion strength using the magnitude of the applied load and the measured area, which would yield more logical results.
7. In Fig.3a, it would be appreciated if the size of the scale bar is indicated.

Reviewer #2 (Remarks to the Author):

This manuscript by Liu et al. titled "A self-powered intracardiac pacemaker," presents a pacemaker that is being recharged by harvesting the mechanical energy of the beating heart using the triboelectric method. The authors designed their device using a leadless pacemaker as a prototype, but instead of a battery, as the power source in commercially available pacemakers, they used a capacitor to store energy and a triboelectric component as the source of energy for recharging the capacitor. Such an approach has been presented earlier in references 32-33. Thus, novelty is limited. Still, this is potentially an exciting technology. However, the data presentation lacks many essential pieces of information, raising concerns about the validity of their conclusions:

1. The manuscript lacks any information regarding the reproducibility of the findings and statistics. How many animals did they use? How did the results compare across animals, time of the experiments, frequency of stimulation, etc? How many measurements of voltage, current, and charge were made? Quantitative results are presented just as numbers without standard deviation or error. What was the data variability?
2. Triboelectric energy harvesting has been demonstrated previously and is reproduced convincingly in a bench test in this paper (Figure 2), but more rigor is needed. It would be helpful to show the correspondence between the current and voltage, recorded simultaneously, perhaps as current-voltage loops on I vs V coordinates. Please present statistics of energy production for different angles of triboelectric compartment vs. vector of movement. What was the movement pattern in vivo? Can you estimate it from two-axis fluoroscopy?
3. The main question remains unanswered in the paper: is a triboelectric energy source capable of producing sufficient energy for long-term pacing? From the text and data, I estimated that it takes about 2.5 hours to charge a capacitor to 3V, but it takes only a couple of minutes to discharge it during pacing (Figure 3E shows a discharge from 3V to about 1.5V in 40 seconds). Please correct me if I am wrong and present quantitative data on the charge time of a capacitor and its discharge below the pacing threshold. By the way, what was the pacing threshold in your system and model? You used 0.5ms pulse width. What was the voltage or current threshold? What was the impedance?
4. I am confused regarding the design of the pacemaker in terms of its attachment to the myocardium. Figure 1 shows four active anchors similar to a commercially available leadless pacemaker Micra from Medtronic. But the fluoroscopy images (Figure 4) and supplementary video clearly show a different attachment method by a screw. How did you attach the pacemaker? And what was the design of the electrode?
5. A critical discrepancy was noted between current and voltage recordings in Figure 4C. How do you explain the lack of synchrony between waveforms of current and waveforms of voltage recorded simultaneously in vivo? These time series appear of different frequencies and lack any correlation between them. How is it possible? Please provide I-V trajectories and quantification in multiple animals and at different times.
6. Figure 5 aims to demonstrate long-term in vivo pacing with the self-charging pacemaker. However, even in these two short traces, the pacing was unreliable. The upper trace of Figure 5F shows pacing at 108 beats per minute, but the capture was unstable. There is a sinus beat after three paced beats. What was the duration of pacing versus sinus rhythm in your experiments? What percent of the time have you succeeded in capture of the heart rhythm by your pacemaker, depending on the stimulation rate? Do I understand correctly that this is not an on-demand pacemaker?

Reviewer #3 (Remarks to the Author):

In the article entitled "A Self powered intracardiac pacemaker" Liu et al describe a novel nano technology in pacemaker development whereby cardiac motion is used to generate energy which the

authors claim his superior to both electromagnetic as well as piezoelectric technology for electricity development former being non compatible with MRI and the latter producing voltage that is not compatible with pacing threshold requirement. This is then converted into electrical energy to enable a 42 mm leadless device measuring 1.75 grams to act as a self powered leadless pacemaker. They conducted invitro and invivo experiments and inserted the device into the right ventricular apex as a leadless pacemaker generating its own energy. Pacemaker is also said to have the capability of recognizing cardiac arrhythmia through sensors at its tip and capable of aborting the arrhythmia. In their experiments conducted in large animal swine model they have demonstrated development of energy over a period of 3 weeks. The authors have provided histopathology at the site of the pacemaker implant in one swine and report no irreversible cardiac damage.

Considering the aging world-wide population particularly in the West, the need for cardiac pacing is going to progressively increase largely for conduction abnormalities but also for other reasons such as heart failure which may benefit from physiologic biventricular pacing. Besides cardiac pacing intracardiac devices, there are other intracardiac devices being used for detection and monitoring of intracardiac pressures as well as arrhythmias and for delivering either mechanical treatment or allowing tailoring of pharmacologic therapy based on the monitoring. Hence the technology proposed by authors is highly desirable and novel.

Currently used right ventricular pacing leads are prone to multiple complications including infection at the pacemaker pocket site (which can also in fact cardiac valves resulting in valve dysfunction), mechanical effects on tricuspid valve leading to tricuspid valve regurgitation sometimes ending in severe right-sided heart failure and significant morbidity and mortality.

While leadless pacemakers have been developed which do not have the complications of pacemaker lead and the pacemaker pocket infection and associated complications as above, these do not have capability of generating electrical power and are difficult to recharge wirelessly due to presence of intracardiac blood, thus requiring replacement which is often difficult besides the devices being expensive.

General comments

- The device length of 42 mm is reasonably large. Please comment if it would not interfere with the papillary muscle and cords and if you had any problem with entanglement of the device with the right ventricular papillary muscle and chordal apparatus.
- The authors have talked about ability of the pacemaker to sense an abnormal electrical impulse and aborting it via electrostatic induction. They demonstrate a PVC and a paced beat after PVC induction. Please clarify if sustained arrhythmia was induced and how and if pacemaker was able to abort it. This section of the paper on aborting arrhythmias is weak.
- Video to shows tensile stress test of the pacemaker: please explain its implications when the pacemaker inside the RV endocardium, in particular its effect on RV endo-myocardial damage. One histology experiment did not show significant damage but was this a consistent finding or in an n of 1.
- Please describe the effect of respiration on charge generation. Respiration may explain the fluctuation in blood flow and electrical signal of the pacemaker. Please comment on effect of respiration in discussion, lines 288-290
- The hook structure at the tip of the self-powered pacemaker unit appears to be different than the screw-in pacemaker leads. How do you ensure that the hook stays in place and is it comparable in strength to the currently available screw-in pacing lead tips. Also please comment on the safety of this method on RV endocardium and myocardium
- Is this pacemaker capable of inducing arrhythmias, did you observe any in your in vivo experiments.
- Where all the experiments conducted in a closed chest swine and how many swine were used.
- Did not see supplementary figures as mentioned in the manuscript line to 17 supplementary figures 7 or supplementary figure 8 line to 239 and supplement figure 9 line 241, figures 10 and 11, lines 246 and 253, figure 14, 274, 16, line 315,
- Echo images in figure 5 b are difficult to discern. It looks like both before and after implantation there is tricuspid regurgitation since the color is shown in blue which means that the flow is going away from the transducer which would generally be from tricuspid regurgitation, please label the

figures to clarify cardiac chambers as well as which views being shown (off axis apical 4?).

- How many experiments were conducted in vivo in the swine model.
- Figure 1 the labeling of RV endocardium appears to be incorrect and appears to be the RV epicardium as the device or the pacemaker capsule is in contact with RV endocardium.
- Typo line 200, sentence needs editing. Line 2019 typo "construction"

Authors' point by point response to the reviewers' comments

Manuscript ID: NCOMMS-23-28297-T

Title: A self-powered intracardiac pacemaker

Our point-by-point responses to reviewers' comments are detailed as follows. Responses are in blue. And the detailed revisions on our manuscript are **highlighted**.

Reviewer#1:

This comprehensive study presents a self-powered intracardiac pacemaker (SICP) that utilizes a nanogenerator to harness biomechanical energy from cardiac motion. The SICP, designed with a capsule structure, can be implanted in the right ventricle using a delivery catheter. It effectively converts cardiac motion energy into electricity, maintaining endocardial pacing function over a three-week follow-up period. The device demonstrates impressive open-circuit voltage and short-circuit current values, showcasing its potential to overcome energy limitations in implantable pacemakers. The SICP's lightweight design, stable energy harvesting performance, and successful testing in swine models highlight its promising role in treating arrhythmia and extending the service life of leadless pacemakers. In the sense that this work represents a significant advancement in the field of implantable bioelectronics, I would recommend this manuscript for possible publication in Nature Communications after minor revision.

Responses:

Thank you for your time and attention to our manuscript. We appreciate your positive comments which encourage us so much. Your detailed and professional critiques and advices are very helpful to us.

Comments:

1. Some important articles about implantable triboelectric nanogenerators and their biomedical applications could be referred to provide readers with more diverse examples of research such as: "doi.org/10.1038/s41467-021-24417-w",

"doi.org/10.1126/science.aan3997", "doi.org/10.1002/smt.202201350", "doi.org/10.1002/adma.202209054". Citing these recent articles can provide supplemental information for the phrase 'to power supply in long-term operation due to battery capacity limitation' in line 60.

Responses:

We appreciate the reviewer very much for this significant suggestion. In this work, we proposed a self-powered intracardiac pacemaker (SICP) with a capsule structure for harvesting biomechanical energy from cardiac motion based on the nanogenerator technology. Before that, some fundamental papers about biomechanical energy harvester also play a key role in solving battery capacity limitations for bioelectronic devices. We greatly respect the contribution of these works to the advancement of this field. We have added these fundamental papers in the reference. Following your constructive suggestion, we carefully checked and modified the references of the manuscript.

Revised in manuscript:

14. Xiao, X. et al. Ultrasound-Driven Injectable and Fully Biodegradable Triboelectric Nanogenerators. *Small Methods*, 2201350 (2023).

15. Meng, X. et al. An Ultrasound-Driven Bioadhesive Triboelectric Nanogenerator for Instant Wound Sealing and Electrically Accelerated Healing in Emergencies. *Advanced Materials* 35, 2209054 (2023).

34 Hinchet, R. et al. Transcutaneous ultrasound energy harvesting using capacitive triboelectric technology. *Science* 365, 491-494 (2019).

35 Ryu, H. et al. Self-rechargeable cardiac pacemaker system with triboelectric nanogenerators. *Nature communications* 12, 4374 (2021).

2. To ensure accurate information, I kindly request a typo correction. In line 156, 'PFEF film' appears to refer to 'PTFE film' in the given context.

Responses:

We sincerely thank you for the correction. We have modified this 'PFEF film' to 'PTFE film'. We are truly sorry for the mistake, and we will pay attention to the

writing of proper nouns.

Revised in manuscript:

...Due to the electret properties of the PTFE film...

3. In Fig.2a, a series of images captured with a high-speed camera showcases the movement process of the POM pellet. It would be helpful to provide information on the total time it took to capture the images as an indication of the time intervals between each photo. This information would contribute to a better understanding of the movement process, considering the driving speed as a factor influencing the POM pellet's movement.

Responses:

We are so grateful for your instructive advice. We employed a high-speed camera to capture pellets movement trajectory, which clearly shows that pellets move sequentially from one end to the other on the side wall under the excitation of the external mechanical motion. According to your suggestion, we have marked the time points on each picture in Fig. 2a based on frame per second of the high-speed camera. This information really helps to better understand the movement process.

Revised in manuscript:

4. In Fig.2d, voltage measurements are presented according to the driving angle of the device. It is observed that all output patterns were driven at the same frequency. It would be beneficial to provide details of the experimental setup used to standardize the applied speed and angle. This information would greatly assist other researchers in conducting follow-up studies.

Responses:

Thank you for your professional suggestion. As we know, the heart contracts and relaxes periodically at a certain frequency. When the device is implanted in the right

ventricle, it is at an angle to the direction of heart motion. Therefore, we systematically evaluated the relationship between the output performance of the device and the tilt angle *in vitro*. In detail, a linear motor was employed to drive the device for keeping the same frequency and speed, which could be controlled by software setting parameters such as displacement, acceleration and velocity. In addition, to create different angles between the device and the direction of motion of the motor, we used 3D printing technology to manufacture four different modules with different angles. We attached the device to different angled modules, which were fixed at the bottom of a linear motor. The device moves up and down with the movement of a linear motor. Through this standardized method, we can achieve different angles between the device and the direction of motion of the motor. We have supplemented a schematic diagram of this experimental procedure. More details of experimental process are show in Figure S3.

Revised in manuscript:

Supplementary Fig.3| Schematic illustration of EHU when working under different tilt angles.

5. According to the manuscript and Fig.2f, a tensile test was conducted to demonstrate the excellent fixation effect of the device. However, peel-off testing is generally considered a more suitable evaluation method for demonstrating the fixation effect. It would be helpful to know the reasons behind conducting the test as described in the manuscript.

Responses:

Thank you for your insightful comments. Peel-off testing is a basic form of mechanical testing that can be performed on a universal testing machine, which is generally considered a suitable evaluation method for demonstrating the fixation effect. Peel-off tests are used to measure the properties of adhesive materials-double faced adhesive tape. We know that peel-off test method has been used in the study of adhesive hydrogel work (Nature, 2019, 575(7781): 169-174; Science, 2023, 381(6658): 686-693; Advanced Functional Materials, 2023: 2303696). In this work, SICP was fixed with the endocardium by hooks. The main factor affecting the fixation effect of the device in the endocardium is the ability of the hooks to resist the tensile strength. Therefore, the tensile test was employed for SICP *in vitro*.

6. Building upon the previous comment, in Fig.2f, only the information on the applied load over time is provided, which seems different from the factors typically associated with demonstrating an excellent fixation effect. It is suggested to calculate values such as adhesion strength using the magnitude of the applied load and the measured area, which would yield more logical results.

Responses:

Thank you for your instructive advice. Using the magnitude of the applied load and the measured area to calculate the bond strength, which is critical for characterizing the properties of materials such as adhered hydrogels. As mentioned in the reply to the previous question, our device is fixed on the endocardium entirely by hooks. Therefore, the main factor affecting the fixation effect of the device in the endocardium is the ability of the hooks to resist the tensile strength. We fixed the device to the isolated heart tissue with four hooks *in vitro*. Tension machine was employed to apply periodic tension. The purpose of this test is to verify the feasibility of device implantation and fixation *in vivo*. The device can withstand a cyclic action of 1.5 N tensile force, indicating that device achieves good fixation in the heart.

7. In Fig.3a, it would be appreciated if the size of the scale bar is indicated.

Responses:

It's very grateful for your professional advices. We have added the size of the scale bar in figure legend.

Revised in manuscript:

...a, Photographs of the integrated PMU&PM (Scale bar = 5 mm)...

Reviewer#2:

This manuscript by Liu et al. titled "A self-powered intracardiac pacemaker," presents a pacemaker that is being recharged by harvesting the mechanical energy of the beating heart using the triboelectric method. The authors designed their device using a leadless pacemaker as a prototype, but instead of a battery, as the power source in commercially available pacemakers, they used a capacitor to store energy and a triboelectric component as the source of energy for recharging the capacitor. Such an approach has been presented earlier in references 32-33. Thus, novelty is limited. Still, this is potentially an exciting technology. However, the data presentation lacks many essential pieces of information, raising concerns about the validity of their conclusions.

Responses:

Thank you for your time and attention to our manuscript. We appreciate your positive comments which encourage us so much. Your detailed and professional critiques and advices are very helpful to us.

This work presents a new methodology of biomechanical energy harvesting from the chambers of the heart with high conversion efficiency, good biocompatibility, and favorable surgery safety through a couple effect of triboelectrification and electrostatic induction, which is fundamentally different from the previously reported biomechanical energy harvesting methods. It is the first experimental demonstration that self-powered intracardiac pacemaker (SICP) has a distinctive capsule shape integrated with a sheath catheter for implanting in the right ventricle of a swine through intravenous. This device can convert cardiac motion energy to electricity effectively. The *in vivo* open circuit voltage and short circuit current of SICP were about 6.0 V and 0.2 μ A, respectively. The leadless structure design of SICP maintains endocardial pacing function during three weeks (21 days) follow-up period *in vivo*. This is the first report of the longest implantation period of a new type of pacemaker in a large animal.

Our study reveals that this device (1.75 g, 1.52 cc) does not affect normal physiological functions of the heart in actual work. In addition, the energy harvesting

unit is mainly constructed of polymer materials, which provides feasibility for SICP to be compatible with MRI examinations during clinical translation applications in the future. This work may promote the development of the next-generation of cardiac pacemakers and implantable bioelectronics.

These key findings together with their application potential can make this paper significant. To better reflect the innovation of this work, we systematically compared the difference between this device and previous work (Table S1).

Table S1: Compared the difference between this device and previous work

Time	Reference	Type	Animal model	Implant location	Intervention operation	Pacing function	MRI-compatible	Size (mm) or Mass	Duration of implantation in vivo	Electrical properties
2010	Advanced materials, 2010, 22(23): 2534-2537	PENG	Rat	Surface of the heart	NO	NO	YES	-(nano wire)	Acute phase Experiment	1 pA, 1 mV
2013	Annals of Biomedical Engineering, 2013, 41(1),131-141	EMG	Sheep	Surface of the heart	NO	NO	NO	-/16.7 g	Acute Phase Experiment	< 1V, 16.7 μW
2014	Advanced materials, 2014, 26(33): 5851-5856	TENG	Rat	Subcutaneous	NO	YES	YES	12 × 12 × 1	Acute Phase Experiment	3.73 V and 0.14 μA
	PNAS, 2014, 111(5): 1927-1932	PENG	Bovine	Surface of the heart	NO	NO	YES	~20 × 60	Acute Phase Experiment	~4 V
2018	Nature communications, 2018, 9(1): 5349	TENG	Rat	Surface of stomach	NO	NO	YES	15 × 20	100 days	< 0.1 V
2019	Nature communications, 2019, 10(1): 1821	TENG	Swine	Surface of the heart	NO	YES	YES	40 × 60 × 1	Acute Phase Experiment	65.2 V, 0.495 μJ
	Acs Nano, 2019, 13(3): 2822-2830	PENG	Swine	Surface of the heart	NO	YES	YES	10 × 40	Acute Phase Experiment	15 μA, < 20 V
2020	Advanced healthcare materials, 2020, 9(11): 2000053	PENG	Swine	Intracardiac	YES	NO	YES	-	Acute Phase Experiment	~ 2 V
	Plos one, 2020, 15(9): e0239667	EMG	Swine	Intracardiac	YES	NO	NO	1.15 cc, 8.01 g	Acute Phase Experiment	< 20 mV
2021	Nature communications, 2021, 12(1): 4374.	TENG	Dog	Subcutaneous	NO	YES	YES	Diameter:~30 cm	24 h	~ 4 V
2023	This work	TENG	Swine	Intracardiac	YES	YES	YES	1.52 cc, 1.75 g	21 days	~6 V, 0.2 μA

1. The manuscript lacks any information regarding the reproducibility of the findings and statistics. How many animals did they use? How did the results compare across animals, time of the experiments, frequency of stimulation, etc? How many measurements of voltage, current, and charge were made? Quantitative results are presented just as numbers without standard deviation or error. What was the data variability?

Responses:

It's very grateful for your professional advises. Performing cardiac intervention experiments on large animal models during the basic research stage poses significant challenges. In this study, a total of eight animals were utilized to conduct the experiments. (Fig. S17). And the overall experimental design idea is gradually deepened. First, we used one swine for inducing the AVB animal model by radiofrequency ablation and exploring the pacing efficiency for PM of SICP in vivo. Then, four swine were employed for evaluating the performance of homemade introducer and dilator advancement, energy harvesting of SICP in vivo, and pacing effect of SICP in acute phase. Finally, three swine were used for evaluating the long-term stability of SICP in vivo. Our data are obtained from multiple experiments. With the continuous improvement of devices and delivery systems, the experimental cycle of this subject has exceeded three years. We performed statistical analysis on both in vivo and in vitro data (Figure 2c, Figure 2d, Figure 3d and supporting information data). The measurement duration/method of voltage, current and charge fully comply with the quantification guidelines and standards of nanogenerators. The weights of the selected animals are all within the same range. However, due to individual variations, there are differences in the intensity and amplitude of heartbeats. As a result, each experiment carries a certain margin of error. So that the in vivo data we present are average values. This work focuses on the feasibility of SICP prototype. As mentioned by the reviewer, to advance the clinical research study of our device, we will conduct statistical research on large sample sizes. This will be a key focus of our subsequent research on this topic.

2. Triboelectric energy harvesting has been demonstrated previously and is reproduced convincingly in a bench test in this paper (Figure 2), but more rigor is needed. It would be helpful to show the correspondence between the current and voltage, recorded simultaneously, perhaps as current-voltage loops on I vs V coordinates. Please present statistics of energy production for different angles of triboelectric compartment vs. vector of movement. What was the movement pattern in vivo? Can you estimate it from two-axis fluoroscopy?

Responses:

We are so grateful for your insightful advice. Open circuit voltage serves as a representative electrical parameter for triboelectric nanogenerators, which can directly reflect the electrical performance of the device. Therefore, we compared the open circuit voltage of EHU at different tilt angles. It is observed that the open circuit voltage of EHU decreases gradually as the angle increases. Building upon the suggestion, we further show the relationship between short-circuit current and different angles, noting similar changing patterns. To calculate the energy production for different angles of triboelectric compartment vs. vector of movement, we examine the relationship between short-circuit charge and different angles (Figure R1a and R1b). The maximum energy output of EHU can be derived by the following equation:

$$E = \bar{P}T = \int_0^T VI dt = \int_{t=0}^{t=T} V dQ = \oint V dQ$$

$$E_{max} = \frac{1}{2} Q_{sc,max} (V_{OC,max} - V_{OC,min})$$

$$E_{max} = \frac{1}{2} Q_{SC,max} \Delta V$$

Here, the average output power \bar{P} was related to the load resistance. E_{max} represents the maximal output energy per cycle. The $Q_{SC,max}$ and ΔV of EHU with different angles were about 21.8 V/6.5 nC (0°), 18.3 V/5.7 nC (15°), 14.5 V/4.6 nC (30°) and 6.85 V/ 1.99 nC (45°), respectively. Therefore, the maximum energy output of EHU with different angles were about 0.071 μJ (0°), 0.052 μJ (15°), 0.033 μJ (30°) and 0.007 μJ (45°), respectively.

Video S5 demonstrates the movement of SICP during the heartbeat, following fixation

under fluoroscopic guidance. As the heart contracts and relaxes, the device's motion pattern includes forward/backward motion and swing (Figure R1c).

Figure R1. **a**, I_{sc} and Q_{sc} of EHU when working under different tilt angles. **b**, Statistical analysis of V_{oc} and I_{sc} of EHU when working under different tilt angles. **c**, Photographs of SICIP movement with heart beat after fixation under fluoroscopic guidance.

3. The main question remains unanswered in the paper: is a triboelectric energy source capable of producing sufficient energy for long-term pacing? From the text and data, I estimated that it takes about 2.5 hours to charge a capacitor to 3 V, but it takes only a couple of minutes to discharge it during pacing (Figure 3E shows a discharge from 3 V to about 1.5 V in 40 seconds). Please correct me if I am wrong and present quantitative data on the charge time of a capacitor and its discharge below the pacing threshold. By the way, what was the pacing threshold in your system and model? You used 0.5 ms pulse width. What was the voltage or current threshold? What was the impedance?

Responses:

Thank you for your professional suggestions. The charge time of a capacitor from 0 V to 3 V is 8734s (~2.43 h). Firstly, it is important to clarify that brady-arrhythmia was intermittently occurred, which means the pacemaker does not operate continuously. The pulse transmitter is triggered to perform electrical stimulation only when a brady-arrhythmia occurs. Under normal circumstances, typically 1-3 electrical pulses are sufficient to correct the heart to a normal heart rate (Response Figure 2, Hesselson, & Aaron, B (2008). Simplified Interpretation of Pacemaker ECGs. DOI:10.1002/9780470695982).

Figure R2. Electrocardiogram of an electrical pulse regulating bradycardia

We have calculated the energy output of SICP during each cardiac motion cycle in supporting information: SICP converts biomechanical energy from cardiac motion to electricity with time-dependent. The average output power \bar{P} was related to the load resistance. The maximum energy output per cycle of SICP can be derived by the following equation:

$$E = \bar{P}T = \int_0^T VI dt = \int_{t=0}^{t=T} V dQ = \oint V dQ$$

$$E_{max} = \frac{1}{2} Q_{sc,max} (V_{OC,max} - V_{OC,min})$$

$$E_{max} = \frac{1}{2} Q_{SC,max} \Delta V$$

Here, E_{max} represents the maximal output energy per cycle. The $Q_{SC,max}$ and ΔV of SICP *in vivo* were about 6.0 V and 8.5 nC, respectively. Therefore, the E_{max} of SICP for per cycle is about 0.026 μ J.

The pacing threshold energy can be derived by the following equation:

$$E_t = \int_0^T V_t \times I dt = V_t \times I \times T = \frac{V_t^2 \times T}{R}$$

Here, E_t represents the pacing threshold energy, V_t is the pacing threshold voltage. R represents the pacing resistance. T stands for stimulus pulse durations.

The mean pacing threshold voltage of SICP is 1.5 V with a pulse width of 0.5 ms, the mean pacing impedance of swine is about 953 Ω (Furrer M, Fuhrer J, Altermatt H J, et al. Surgical endoscopy, 1997, 11(12): 1167-1170.). Therefore, the mean pacing threshold energy of SICP is 1.18 μJ in animal experiment. On the other hand, Ritter, P. et al. reported early performance clinical test of a miniaturized leadless cardiac pacemaker - Medtronic's Micra TPS (European heart journal 2015, 36, 2510-2519). The mean pacing capture threshold at the 3-month visit for the 60 patients measured with a pulse width of 0.24 ms was 0.51 V (95% CI, 0.45–0.56; $P < 0.0001$), meeting the efficacy objective. Among these 60 patients, the mean electrical values for R-wave sensing amplitude, pacing impedance, and pacing capture threshold at a pulse width of 0.24 ms were as follows respectively: 11.7 \pm 4.5 mV, 719 \pm 226 ohm, 0.57 \pm 0.31 V at implant, 15.6 \pm 4.8 mV, 662 \pm 133 ohm, 0.48 \pm 0.21 V at 1-month, and 16.1 \pm 5.2 mV, 651 \pm 130 ohm, 0.51 \pm 0.22 V at 3-months.

$$E_{\text{maximum pacing threshold}} = (0.88 \text{ V})^2 \times 0.24 \text{ ms} \div 493 \Omega = 0.377 \mu\text{J}$$

$$E_{\text{meanmum pacing threshold}} = (0.51 \text{ V})^2 \times 0.24 \text{ ms} \div 719 \Omega = 0.087 \mu\text{J}$$

$$E_{\text{minimum pacing threshold}} = (0.26 \text{ V})^2 \times 0.24 \text{ ms} \div 945 \Omega = 0.017 \mu\text{J}$$

Therefore, based on the rough calculation we can draw the following conclusion:

$$E_{\text{minimum pacing threshold}} < E_{\text{max}} = 0.026 \mu\text{J} = 1/3.3 E_{\text{meanmum pacing threshold}} = 1/14.5$$

$$E_{\text{maximum pacing threshold}}$$

Theoretically, it means that the energy harvested by SICP from four heartbeats can satisfy the leadless cardiac pacemaker for one pacing. Based on the working frequency of the pacemaker and the actual needs of each pacing electrical pulse, the triboelectric energy source of SICP is capable to produce sufficient energy for long-term pacing. Furthermore, with the in-depth follow-up research and the improvement

of the efficiency of the EHU, we believe that the energy collected by SICP from one heartbeat can fully satisfy the leadless pacemaker for one pacing.

Revised in manuscript:

...The biomechanical energy harvested of SICP from each cardiac cycle is about 0.026 μJ . Theoretically, it means that the energy harvested by SICP from four heartbeats can satisfy the leadless cardiac pacemaker for one pacing (Supplementary Note 1). Maximal Power output of SICP is about 0.039 μW (Supplementary Note 2). Based on the working frequency of the pacemaker and the actual needs of each pacing electrical pulse, the triboelectric energy source of SICP is capable to produce sufficient energy for long-term pacing. Furthermore, with the in-depth follow-up research and the improvement of the efficiency of the EHU, we believe that the energy collected by SICP from one heartbeat can fully satisfy the leadless pacemaker for one pacing...

Revised Supplementary information:

The pacing threshold energy can be derived by the following equation:

$$E_t = \int_0^T V_t \times I dt = V_t \times I \times T = \frac{V_t^2 \times T}{R}$$

Here, E_t represents the pacing threshold energy, V_t is the pacing threshold voltage. R represents the pacing resistance. T stands for stimulus pulse durations.

The mean pacing threshold voltage of SICP is 1.5 V with a pulse width of 0.5 ms, the mean pacing impedance of swine is about 953 Ω^2 . Therefore, the mean pacing threshold energy of SICP is 1.18 μJ in animal experiment. On the other hand, Ritter, P. et al. reported early performance clinical test of a miniaturized leadless cardiac pacemaker - Medtronic's Micra TPS³. The mean pacing capture threshold at the 3-month visit for the 60 patients measured with a pulse width of 0.24 ms was 0.51 V (95% CI, 0.45–0.56; $P < 0.0001$), meeting the efficacy objective. Among these 60 patients, the mean electrical values for R-wave sensing amplitude, pacing impedance, and pacing capture threshold at a pulse width of 0.24 ms were as follows respectively:

11.7±4.5 mV, 719 ± 226 ohm, 0.57±0.31 V at implant, 15.6±4.8 mV, 662 ± 133 ohm, 0.48 ± 0.21 V at 1-month, and 16.1 ± 5.2 mV, 651 ± 130 ohm, 0.51 ± 0.22 V at 3-months.

$$E_{\text{maximum pacing threshold}} = (0.88 \text{ V})^2 \times 0.24 \text{ ms} \div 493 \Omega = 0.377 \mu\text{J}$$

$$E_{\text{meanmum pacing threshold}} = (0.51 \text{ V})^2 \times 0.24 \text{ ms} \div 719 \Omega = 0.087 \mu\text{J}$$

$$E_{\text{minimum pacing threshold}} = (0.26 \text{ V})^2 \times 0.24 \text{ ms} \div 945 \Omega = 0.017 \mu\text{J}$$

Therefore, based on the rough calculation we can draw the following conclusion:

$$E_{\text{minimum pacing threshold}} < E_{\text{max}} = 0.026 \mu\text{J} = 1/3.3 E_{\text{meanmum pacing threshold}} = 1/14.5$$

$$E_{\text{maximum pacing threshold}}$$

2. *Furrer, M. et al. VATS-guided epicardial pacemaker implantation: hand-sutured fixation of atrioventricular leads in an experimental setting. Surgical endoscopy 11, 1167-1170 (1997).*

3. *Ritter, P. et al. Early performance of a miniaturized leadless cardiac pacemaker: the Micra Transcatheter Pacing Study. European heart journal 36, 2510-2519 (2015)*

4. I am confused regarding the design of the pacemaker in terms of its attachment to the myocardium. Figure 1 shows four active anchors similar to a commercially available leadless pacemaker Micra from Medtronic. But the fluoroscopy images (Figure 4) and supplementary video clearly show a different attachment method by a screw. How did you attach the pacemaker? And what was the design of the electrode?

Responses:

Thank you for your insightful comments. As described in the Methods section, we utilized nickel alloy wires, which were bent into an arch shape with a diameter of 0.3 mm and a length of 10 mm. These wires were threaded through holes in the bottom of the SICP to serve as hooks for anchoring in the myocardial wall. The spiral platinum-iridium alloy was attached to the bottom of SICP as cathode, which also creates a good connection with the endocardium. Both the four active anchors and the spiral

electrodes have fixed roles in ensuring the secure fixation of the device. We proposed this design idea to enhance the device's stability and improve its overall fixation. It is worth noting that the imaging visibility of materials under X-rays depends on various factors, including material properties (such as X-ray absorption rate and thickness) and the positioning of the animal's body. Due to the small thickness of the active anchor (only 0.3 mm) and the fact that the entire device is located inside the heart, the imaging effects *in vivo* are compromised. As a result, the fixtures are challenging to visualize in the fluoroscopy images (Figure 4) and supplementary video. The device implantation surgery was performed by a pacemaker clinical specialist, who possesses the expertise to assess the firmness of the device based on its positioning and swing state. During the experiment, we also captured fluoroscopy image of SICP in the Left Anterior Oblique position (Figure R3a) and physical pictures of the anatomy, where the hook can be vaguely observed (Figure R3b).

Figure R3. **a**, fluoroscopy images of SICP in different detection positions. **b**, Physical pictures of the anatomy of SICP.

5. A critical discrepancy was noted between current and voltage recordings in Figure 4C. How do you explain the lack of synchrony between waveforms of current and waveforms of voltage recorded simultaneously *in vivo*? These time series appear of different frequencies and lack any correlation between them. How is it possible? Please provide I-V trajectories and quantification in multiple animals and at different times.

Responses:

Thank you for your professional suggestions. The current and voltage recordings

presented in Figure 4C were indeed obtained by the same SICP implanted in the same animal. After the device was implanted into the ventricle through minimally invasive intervention, we characterized the voltage and current of the device by an electrometer and oscilloscope. During the experiment, we took into consideration that simultaneously measuring two electrical parameters (the voltage and current) could potentially affect the accuracy and stability of data measurement. Therefore, the voltage and current of the device were tested sequentially. It is worth mentioning that in the field of nanogenerators, voltage and current characterization commonly employ this approach.

Due to the periodic contraction and relaxation of the heart, there is free movement of globules inside SICP, leading to some volatility in the electrical signal. We speculate that fluctuations in the electrical signal may also be affected by the blood flow. In addition, due to the signal acquisition rate of the measurement equipment in vivo, some small voltage signals may not be captured. Therefore, the discrepancy in frequencies observed between the voltage and current data can be attributed to the testing principles of these two electrical quantities and the structure of our device. Following the suggestion, we also provide I-V trajectories and quantification in the animal at different times (Figure R4). The results also show that the voltage and current waveforms exhibit similar characteristics.

Figure R4. I-V trajectories and quantification in the animal at different times.

6. Figure 5 aims to demonstrate long-term in vivo pacing with the self-charging pacemaker. However, even in these two short traces, the pacing was unreliable. The upper trace of Figure 5F shows pacing at 108 beats per minute, but the capture was

unstable. There is a sinus beat after three paced beats. What was the duration of pacing versus sinus rhythm in your experiments? What per cent of the time have you succeeded in capture of the heart rhythm by your pacemaker, depending on the stimulation rate? Do I understand correctly that this is not an on-demand pacemaker?

Response:

Thank you for your insightful comments. The ECG of a unipolar pacemaker usually displays a signal perpendicular to the ECG baseline before each cardiac cycle, called a spike signal, which is the pacing maker in ECG signal of the pacemaker. However, bipolar pacemaker spike signals are generally subtle and often difficult to identify. Therefore, changes in heart rate reflected by ECG are key parameters for evaluating whether bipolar pacing is reliable or not. The pacing mode of our device is VOO (ventricular asynchronous pacing), which would deliver the electrical impulse at a regular interval. In addition, our device is bipolar pacing pacemaker. In Figure 5F, during the fourth electrical impulse generated by the device, the ventricular myocardium was activated simultaneously with the stimulus originating from the sinus node, and therefore generating the fusion wave. This phenomenon is considered normal in the VOO pacing mode. The morphology of the fourth paced QRS complex differs from other paced QRS complexes (Figure R5, Hesselson, & Aaron, B (2008). Simplified Interpretation of Pacemaker ECGs. DOI:10.1002/9780470695982). Therefore, the presence of the fourth QRS complex indicated successful pacing capture by the SICP. We observed stable working of the SICP throughout the experiment. It is important to note that the design of the device in this study aimed to provide a novel approach for sustained energy supply, and we acknowledge that it was not specifically designed as an on-demand pacemaker. However, based on our current research, we aspire to enhance the SICP in the future to enable on-demand pacing functionality.

Figure R5. Contrast electrocardiogram of fusion wave.

Reviewer#3:

In the article entitled: "A Self-powered intracardiac pacemaker" Liu et al describe a novel nano technology in pacemaker development whereby cardiac motion is used to generate energy which the authors claim is superior to both electromagnetic as well as piezoelectric technology for electricity development former being non compatible with MRI and the latter producing voltage that is not compatible with pacing threshold requirement. This is then converted into electrical energy to enable a 42 mm leadless device measuring 1.75 grams to act as a self-powered leadless pacemaker. They conducted invitro and in vivo experiments and inserted the device into the right ventricular apex as a leadless pacemaker generating its own energy. Pacemaker is also said to have the capability of recognizing cardiac arrhythmia through sensors at its tip and capable of aborting the arrhythmia.

In their experiments conducted in large animal swine model they have demonstrated development of energy over a period of 3 weeks. The authors have provided histopathology at the site of the pacemaker implant in one swine and report no irreversible cardiac damage.

Considering the aging world-wide population particularly in the West, the need for cardiac pacing is going to progressively increase largely for conduction abnormalities but also for other reasons such as heart failure which may benefit from physiologic biventricular pacing. Besides cardiac pacing intracardiac devices, there are other intracardiac devices being used for detection and monitoring of intracardiac pressures as well as arrhythmias and for delivering either mechanical treatment or allowing tailoring of pharmacologic therapy based on the monitoring. Hence the technology proposed by authors is highly desirable and novel.

Currently used right ventricular pacing leads are prone to multiple complications including infection at the pacemaker pocket site (which can also in fact cardiac valves resulting in valve dysfunction), mechanical effects on tricuspid valve leading to tricuspid valve regurgitation sometimes ending in severe right-sided heart failure and significant morbidity and mortality.

While leadless pacemakers have been developed which do not have the complications

of pacemaker lead and the pacemaker pocket infection and associated complications as above, these do not have capability of generating electrical power and are difficult to recharge wirelessly due to presence of intracardiac blood, thus requiring replacement which is often difficult besides the devices being expensive.

Responses:

Thank you for your time and attention to our manuscript. We appreciate your positive comments which encourage us so much. Your detailed and professional critiques and advices are very helpful to us.

General comments

1. The device length of 42 mm is reasonably large. Please comment if it would not interfere with the papillary muscle and cords and if you had any problem with entanglement of the device with the right ventricular papillary muscle and chordal apparatus.

Responses:

Thank you for your insightful comments. The length of Nanostim LCP leadless pacemaker is 42 mm. The latest generation of leadless pacemaker (model: AVEIR™VR) was recently approved by FDA in the year of 2023, with a length of 38 mm. The device in this study shares almost the same length to AVEIR™VR and Nanostim LCP (Supplementary Table 1). Such length design of the device is reasonable and acceptable.

Moreover, the experiment conducted on 8 swine successfully demonstrated the device's safe delivery to the right ventricle (RV). The tines of the device could hook into the RV trabeculae and anchor the device inside the RV, while the body and the tail of the device remained free from entanglement with the papillary muscle and chordal apparatus, as confirmed during the post-mortem examinations. Additionally, there were no complications such as cardiac perforation or thromboembolism during the implantation procedure.

Nonetheless, our commitment to enhancing the energy harvesting efficacy of the device remains unwavering in future endeavors. We are dedicated to achieving

smaller dimensions for the next-generation products, drawing it closer to the smallest standalone system.

2. The authors have talked about ability of the pacemaker to sense an abnormal electrical impulse and aborting it via electrostatic induction. They demonstrate a PVC and a paced beat after PVC induction. Please clarify if sustained arrhythmia was induced and how and if pacemaker was able to abort it. This section of the paper on aborting arrhythmias is weak.

Responses:

Thank you for your constructive comments. We acknowledged that there were inaccuracies in this section. The sentence in Page 18 Line 305-306: "when SICP was in operation, premature ventricular depolarization was induced by the electrical pulse stimulus.". We incorrectly referred to the paced QRS complex as premature ventricular depolarization-PVC. Actually, our intention was to demonstrate that when SICP was in operation, the premature paced QRS complex was induced by the electrical pulse stimulus. The ECG showed regular paced QRS complex occurrence, indicating that the ventricle was effectively paced by SICP. We have revised these inaccuracy sentences as we mentioned.

Revised manuscript:

...When SICP was in operation, the premature paced QRS complex was induced by the electrical pulse stimulus (Supplementary Video 8). The ECG showed regular paced QRS complex occurrence ahead of P wave (atrial contraction), indicating that the ventricle was effectively captured by SICP...

3. Video to shows tensile stress test of the pacemaker: please explain its implications when the pacemaker inside the RV endocardium, in particular its effect on RV endomyocardial damage. One histology experiment did not show significant damage but was this a consistent finding or in an n of 1.

Responses:

Thank you for your insightful comments. The *in vitro* tensile stress test of the device,

as presented in Supplementary Video 2, was aimed to assess the fixation status between the hook structure of the device and the RV myocardium. The tensile stress test confirmed the stability of hook structure with RV endocardium. The tine structure of the device bears resemblance to the Micra™ TPS. Furthermore, due to the design of our materials and structure, the device has a low mass of only 1.75 g (<Micra™ TPS, 2 g). This low mass results in minimal interaction forces between the device and the endocardium during cardiac contraction and relaxation. While we acknowledged that this fixation mechanism could potentially result in localized damage to the endocardial myocardium, it is important to note that this damage is confined to a confined area. This confinement has been substantiated by the histology experiment depicted in Figure 5g and 5h. And there was no serious myocardial injury such as cardiac perforation or cardiac rupture throughout both the acute and chronic phase of the study. The histology experiment on all 8 swine did not show significant damage in RV endo-myocardium.

4. Please describe the effect of respiration on charge generation. Respiration may explain the fluctuation in blood flow and electrical signal of the pacemaker. Please comment on effect of respiration in discussion, lines 288-290.

Responses:

Thank you for your constructive comments. In this study, the device was implanted at RV apex through vascular intervention, which could anchor to the myocardial tissue with its helix and hooks of the tip. This device moves back and forth with the cardiac contraction and relaxation, thus harvesting biomechanical energy based on the triboelectric effect. Therefore, the generation of electrical energy during this process is primarily associated with strength and frequency of cardiac contraction. During the experiment, the animals were under anesthesia and on a ventilator while the device was implanted and electrical characterization was performed. As a result, the state of breathing remains unchanged. The electrical signal characteristics of the device *in vivo* shown in Figure 4c, we speculate that fluctuations in the electrical signal may also be affected by the blood flow.

We concur your suggestions. The respiration could also influence the myocardial contractility, subsequently affect the energy harvesting of the device under daily condition. Respiration changes the pressure gradient between the intrathoracic and extrathoracic veins. Hence respiration serves as an auxiliary pump, which may alter the mean level of myocardial contractility and cardiac output, and may influence charge generation of the device during the various phases of the respiratory cycle. Therefore, we have commented the respiration effect in discussion part as your suggested.

Revised manuscript:

...Actually, cardiac contraction intensity is also affected by respiratory status and exercise. Therefore, appropriately enhancing cardiac functional status may be beneficial to improving energy harvesting efficiency...

5. The hook structure at the tip of the self-powered pacemaker unit appears to be different than the screw-in pacemaker leads. How do you ensure that the hook stays in place and is it comparable in strength to the currently available screw-in pacing lead tips. Also please comment on the safety of this method on RV endocardium and myocardium

Responses:

We are so grateful for your instructive advice. The leadless device investigated in this study does not rely on conventional screw-in fixation mechanism employed by traditional pacing lead. Instead, its tine structure of the device draws inspiration primarily from the representative leadless pacemaker, the Micra™ TPS. Compared with TPS, we incorporated additional helix design of the screw-in pacemaker leads (Figure 1b and 1e) based on hooks of the tip. This further ensured the stability and reliability of the device after implanting at the RV apex, thereby guaranteeing the efficacy of energy collection. As illustrated in Figure 4a, we successfully delivered the device via intravenous route using the delivery sheath. The device traversed the tricuspid valve annulus and reached the RV apex, where the tines could securely hook into the RV trabeculae. Notably, there were no differences between the hooks and

screw-in pacing lead tips in terms of stability and fixation strength.

Supplementary Figure 3 showed SICP fixed on the endocardium of the right ventricle in an isolated heart. Furthermore, Supplementary Figure 13 demonstrated the surface of the delivery sheath is exceptionally smooth, mitigating any potential damage to the RV endocardium and myocardium. Importantly, throughout the delivery procedure, the device remains within the protective confines of the sheath, and the tines remain enclosed. The device is deployed by experienced electrophysiologist once the target zone is reached. This delivery methodology closely mirrors the commercial leadless pacemaker, Micra™ TPS. There was no serious myocardial injury such as cardiac perforation, cardiac rupture throughout both the acute and chronic phase of the study. Consequently, we consider that the methodology employed in this study is both safe and effective.

6. Is this pacemaker capable of inducing arrhythmias, did you observe any in your in vivo experiments.

Responses:

Thank you for your insightful comments. We observed non-sustained ventricular tachyarrhythmia during the implantation procedure, which was the normal response of myocardium tissue to the contact with the device. These arrhythmias were transient and would cease once the stimulus ceased. It is worth noting that various devices that come into contact with the myocardium, such as pacing leads or closure devices, can induce arrhythmias during the implantation procedure. However, once the device firmly attached to the myocardium, there would be no stimulus of contact, thus preventing the induction of new arrhythmia episodes. As depicted in Figure 3h, as well as Figure 5c, 5d, 5e, and 5f, the device in this study did not induce arrhythmias after implantation. Among the 8 animals included in our study, no stances of device-related arrhythmias were observed.

7. Where all the experiments conducted in a closed chest swine and how many swine were used.

Response:

All animal experiments were conducted using minimally invasive intervention, eliminating the need for open-chest procedures. Performing cardiac intervention experiments on large animal models during the basic research stage poses significant challenges. In this study, a total of 8 animals were utilized to conduct the experiments. (Fig. S17). And the overall experimental design idea is gradually deepened. First, we used one swine for inducing the AVB animal model by radiofrequency ablation and exploring the pacing efficiency for PM of SICP in vivo. Then, four swine were employed for evaluating the performance of homemade introducer and dilator advancement, energy harvesting of SICP in vivo, and pacing effect of SICP in acute phase. Finally, three swine were used for evaluating the long-term stability of SICP in vivo. Our data are obtained from multiple experiments.

8. Did not see supplementary figures as mentioned in the manuscript line to 17 supplementary figures 7 or supplementary figure 8 line to 239 and supplement figure 9 line 241, figures 10 and 11, lines 246 and 253, figure 14, 274, 16, line 315,

Response:

Thank you for your instructive advice. The figures you mentioned are not well displayed in the supplementary materials, which may be related to the submission system. We will further check and verify these figures when submitting the revised manuscript this time.

9. Echo images in figure 5 b are difficult to discern. It looks like both before and after implantation there is tricuspid regurgitation since the color is shown in blue which means that the flow is going away from the transducer which would generally be from tricuspid regurgitation, please label the figures to clarify cardiac chambers as well as which views being shown (off axis apical 4?)

Response:

Thank you for your professional suggestion. Given the distinctive anatomical structure of the swine thorax, the supine positioning during anesthesia for the

procedure, and the inherent constraints of acoustic windows, the echocardiographic imaging resolution remains suboptimal. That's the reason why the Echo images were difficult to discern. Nevertheless, the results of our study demonstrated that there was no significant worsening of tricuspid regurgitation. This showed that the device did not have significant impact on the structure and the function of heart. We have labelled the figure5b to clarify cardiac chambers as your suggested.

Revised in manuscript:

10. How many experiments were conducted *in vivo* in the swine model.

Response:

A total of 8 animals were included in the study. The details of the experiment were summarized in Supplementary Fig 17.

11. Figure 1 the labeling of RV endocardium appears to be incorrect and appears to be the RV epicardium as the device or the pacemaker capsule is in contact with RV endocardium.

Response:

Thank you for your constructive comments. We have revised the Figure 1a as your suggested.

Revised in manuscript:

12. Typo line 200, sentence needs editing. Line 2019 typo “construction”

Response:

We concurred your comments. we have revised the sentence as your suggested.

Revised in manuscript:

...Before and after **atrioventricular node ablation** in swine model...

REVIEWERS' COMMENTS

Reviewer #1 (Remarks to the Author):

I have carefully reviewed the responses provided by the author and believe that the reviewers' comments have been appropriately addressed in the manuscript and supplementary materials. Thank you for your thorough and thoughtful responses. I think publishing your manuscript in Nature Communications is possible without further revisions.

Reviewer #2 (Remarks to the Author):

I am generally satisfied with the responses. The presented technology is an improvement.

However, the authors should engage an expert in cardiac pacing to help with their terminology. Moreover, the presented technology is still incapable of providing long-term constant pacing. This should be acknowledged in the limitations. Only intermittent pacing is possible due to the small amount of energy harvested by this method versus that required for constant pacing.

Reviewer #3 (Remarks to the Author):

I have reviewed the detailed responses the authors have provided to the 3 reviewers. From clinical standpoint I am satisfied with their responses and have no more comments.

Authors' point by point response to the reviewers' comments

Manuscript ID: NCOMMS-23-28297-B

Title: A self-powered intracardiac pacemaker

Our point-by-point responses to reviewers' comments are detailed as follows. Responses are in blue. And the detailed revisions on our manuscript are **highlighted**.

Reviewer#1:

I have carefully reviewed the responses provided by the author and believe that the reviewers' comments have been appropriately addressed in the manuscript and supplementary materials. Thank you for your thorough and thoughtful responses. I think publishing your manuscript in Nature Communications is possible without further revisions.

Responses:

Thank you again for your time and attention to our manuscript. We appreciate your positive comments which encourage us so much. Your detailed and professional critiques and advices are very helpful to us.

Reviewer#2:

I am generally satisfied with the responses. The presented technology is an improvement.

However, the authors should engage an expert in cardiac pacing to help with their terminology. Moreover, the presented technology is still incapable of providing long-term constant pacing. This should be acknowledged in the limitations. Only intermittent pacing is possible due to the small amount of energy harvested by this method versus that required for constant pacing.

Responses:

Thank you again for your time and attention to our manuscript. We appreciate your positive comments which encourage us so much. Your detailed and professional critiques and advices are very helpful to us.

Prof. Hua and Dr. Hu as experts in cardiac pacing from department of cardiology, the cardiac arrhythmia center, state key laboratory of cardiovascular disease, national clinical research center of cardiovascular diseases, Fuwai hospital, are the main authors of this manuscript. Based on your suggestions, other experts in cardiac pacing have been invited to work with them to check and revise terminology.

The biomechanical energy harvested of SICP from each cardiac cycle is about 0.026 μJ (Supplementary Note 1). Maximal Power output of SICP is about 0.039 μW (Supplementary Note 2). Theoretically, it means that the energy harvested by SICP from four heartbeats will be higher than the pacing threshold energy of commercial leadless cardiac pacemaker (Supplementary Note 1). Three weeks after the operation, the experimental animals maintained a normal survival state, and the device exhibited excellent output performance. Large animal experimental models effectively simulate clinical applications and may provide more valuable and comparable results. Nonetheless, we agree with you that SICP has certain limitations in long-term constant pacing on clinical criterion, which requires a large number of long-term animal experiments for further research. These contents are also the focus of our next research. Based on your suggestions, we have modified the discussion section of the manuscript.

Revised in manuscript:

...to harvest biomechanical energy from **cardiac motion** for powering the pacemaker module...

...thereby preventing the **perioperative** risk caused by the replacement of devices due to energy **depletion**...

...to ensure the **cardiac physiological activity**, the overall device...

...Minimally invasive **intervention** with delivery technology...

~~...Based on the working frequency of the pacemaker and the actual needs of each pacing electrical pulse, the triboelectric energy source of SICP is capable to produce sufficient energy for long-term pacing...~~

...Although SICP has certain limitations in long-term constant pacing on clinical criterion, this work provides a proof-of-concept demonstration for the next generation pacemaker, and will facilitate the upgrade of existing commercial leadless pacemakers (Table S1). Furthermore, with the in-depth follow-up research and the improvement of the efficiency of the EHU, we believe that the energy collected by SICP from one heartbeat can fully satisfy the leadless pacemaker for one pacing...

Reviewer#3:

I have reviewed the detailed responses the authors have provided to the 3 reviewers. From clinical standpoint I am satisfied with their responses and have no more comments.

Responses:

Thank you again for your time and attention to our manuscript. We appreciate your positive comments which encourage us so much. Your detailed and professional critiques and advices are very helpful to us.